# Network-based clustering unveils interconnected landscapes of genomic and clinical features across myeloid malignancies

Fritz Bayer[1,2], Marco Roncador[1,2], Giusi Moffa [3], Kiyomi Morita[4], Koichi Takahashi [4,5], Niko Beerenwinkel [1,2] & Jack Kuipers [1,2] ✉

Myeloid malignancies exhibit considerable heterogeneity with overlapping clinical and genetic features among subtypes. We present a data-driven approach that integrates mutational features and clinical covariates at diagnosis within networks of their probabilistic relationships, enabling the discovery of patient subgroups. A key strength is its ability to include presumed causal directions in the edges linking clinical and mutational features, and account for them aptly in the clustering. In a cohort of 1323 patients, we identify subgroups that outperform established risk classifications in prognostic accuracy. Our approach generalises well to unseen cohorts with classification based on our subgroups similarly offering advantages in predicting prognosis. Our findings suggest that mutational patterns are often shared across myeloid malignancies, with distinct subtypes potentially representing evolutionary stages en route to leukemia. With pancancer TCGA data, we observe that our modelling framework extends naturally to other cancer types while still offering improvements in subgroup discovery.

Acute myeloid leukemia (AML), myelodysplastic syndromes (MDS), chronic myelomonocytic leukemia (CMML), and myeloproliferative neoplasms (MPN) are heterogeneous groups of myeloid malignancies that share overlapping clinical and genetic features[1,2]. Patients with MDS, CMML and MPN are at risk of progressing to advanced leukemic diseases, such as AML[3,4]. This transformation highlights the interconnectedness and the relevance of finding clinically relevant subgroups amongst myeloid malignancies.

Current classifications of myeloid malignancies at diagnosis are primarily dominated by clinical covariates. Although guided by threshold values (e.g., blast values being below or above a certain threshold) of these clinical variables, the classifications have changed significantly over the years and are increasingly driven by mutational and cytogenetic features[5–9]. The presence and interaction of these genetic changes can influence the prognosis and treatment strategy[10–14].

Despite these classification refinements, there remains considerable heterogeneity in clinical features and treatment response within cancer types and cancer subtypes. The task of defining patient subgroups based on genomic profiles is challenging due to the vast number of driver mutations and the multitude of their possible combinations. Consequently, common classifications rely on the presence or absence of a small number of driver mutations, which may not fully encompass the complexity of genomic landscapes. Likewise, the sheer volume of potential combinations of mutations and clinical covariate features make integrated classification even more challenging, resulting in increasingly complex classification tables that combine ranges of clinical covariates with specific mutations and cytogenetic factors[9]. Such complexity restricts the classification to considering only a small number of covariates and genetic features. Understanding the intricate mutational landscape of myeloid malignancies, and their interplay

[1]Department of Biosystems Science and Engineering, ETH Zurich, Schanzenstrasse 44, 4056 Basel, Switzerland. [2]SIB Swiss Institute of Bioinformatics, Schanzenstrasse 44, 4056 Basel, Switzerland. [3]Department of Mathematics and Computer Science, University of Basel, Basel, Switzerland. [4]Department of Leukemia, The University of Texas MD Anderson Cancer Center, Houston, TX, USA. [5]Department of Genomic Medicine, The University of Texas MD Anderson Cancer Center, Houston, TX, USA. ✉e-mail: jack.kuipers@bsse.ethz.ch

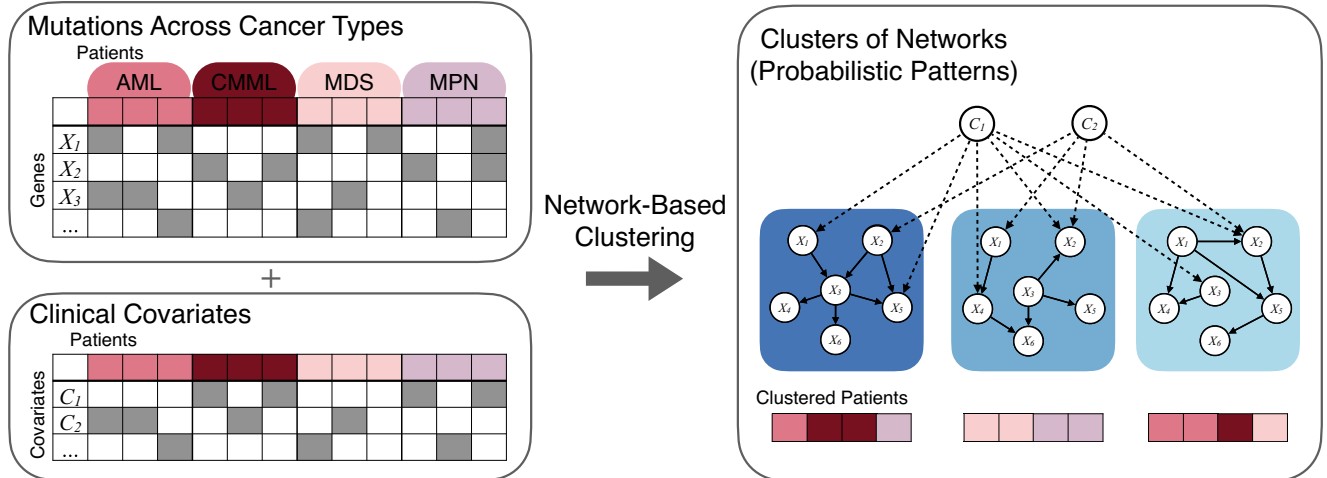

**Fig. 1 | Overview of the network-based clustering.** The data of patients (left panels) across different cancer types (colouring on the left) include mutational profiles ($X_1$, $X_2$, ..., top left) and clinical covariates at diagnosis ($C_1$, $C_2$, ..., bottom left). We model the dependencies amongst the variables in the data with probabilistic network models (right) and use these to cluster the patients into subgroups based on their mutation profiles and distinct probabilistic relationships between the mutations (solid edges, right). Since mutations may depend on the clinical covariates, we account for these dependencies in the network modelling (dashed edges, top right) allowing us to adjust for them in the cluster assignments. The result of the network-based clustering are different network models (shades of blue, right panel) and the assignment of patients to the clusters (bottom right). AML acute myeloid leukemia, CMML chronic myelomonocytic leukemia, MDS myelodysplastic syndromes, MPN myeloproliferative neoplasms.

with other clinical values, is however essential for refining patient stratification, enhancing targeted therapeutic strategies, and improving clinical outcomes.

Leveraging a data-driven approach, previous studies have clustered the mutational and cytogenetic profiles of myeloid malignancies to derive risk scores in AML[10,13,15,16], MDS[14,17,18], CMML[19], and MPS[20,21]. Outside of myeloid malignancies, network-based approaches have been employed to effectively condense heterogeneous genomic profiles for prediction and patient stratification[22–25].

However, most previous clustering strategies did not integrate clinical covariates, such as blood and bone marrow values and demographic factors, into their clustering approaches, despite their established relevance for prognosis and clinical decision-making[26–29]. For instance, a previous study focused exclusively on genomic profiles to cluster MDS patients with secondary AML patients[14]. While insightful, as the authors acknowledged, this approach was unable to integrate clinical data for more nuanced patient stratification. Similarly, in another study encompassing patients diagnosed with AML[13], both mutational and clinical data were considered; however, the authors applied traditional clustering algorithms that do not distinguish between the different types of data. This underscores a need for advancements in methodologies to identify patient clusters through an integrative and comprehensive approach.

Here, we develop a network-based method to integrate both genomic and clinical features at diagnosis, accounting for their different roles in cancer progression and prognosis. Throughout, we include only baseline covariates among the clinical features and exclude survival information, which we instead predict to validate our approach. We benchmark the approach using simulated data and by predicting survival for a large-scale pan-cancer dataset encompassing 8085 patients. Then, we apply the method to myeloid malignancies to understand the interplay between the mutational, cytogenetic, and clinical information in their development. While previous studies have typically clustered AML, MDS, CMML, and MPS individually, our study analyses these malignancies collectively to uncover shared, de novo cancer subgroups across the different cancer types. Specifically, we analyse a pan-myeloid dataset including the mutational profiles and clinical covariates of 1323 patients diagnosed with AML, MDS, CMML, and MPN. By integrating mutational and clinical covariate data within

networks of their probabilistic relationships (Fig. 1), we learn cluster assignments that encompass genomic profiles and clinical covariates holistically, focusing on mutational patterns rather than individual features alone.

## Results
### Analysis Overview
Figure 1 provides an illustration of our clustering methodology, where genomic and clinical data are integrated in networks of their probabilistic relationships. This method allows us to delineate cluster-specific probabilistic associations among genetic mutations and clinical covariates, facilitating a more nuanced understanding of unique patterns within specific patient groups. To avoid learning the clusters based on demographic covariates such as age and sex, we adjust for their effect on the cluster assignment by exploiting causal relationships among the variables (Methods).

We applied our covariate-adjusted network clustering (CANclust) method to a pan-myeloid dataset comprising the mutational and clinical data of 1323 patients with diagnoses of AML, MDS, CMML, and MPN (Supplementary Tables S1 and S2). Our analysis not only clusters samples in an unsupervised manner but also deduces cancer-type-specific networks within the myeloid malignancies, enabling us to explore distinct mutational interactions.

To validate the performance of our method, we conducted comprehensive benchmark tests. These include simulations for a range of scenarios as well as application to a large-scale dataset from The Cancer Genome Atlas (TCGA)[30], featuring the mutational profiles and clinical information of 8085 patients across 22 different cancer types.

### Mutational Patterns within Myeloid Malignancies
Investigating the mutational patterns within myeloid malignancies, we first learned the cancer type-specific networks for AML, CMML, MDS, and MPN by individually applying Bayesian network structure learning to each cancer type-specific dataset within a cohort of 1323 patients (Fig. 2 and Supplementary Fig. S1).

In AML, we observed that the most frequent mutations were *NPM1*, *FLT3*, and *DNMT3A*, with *NPM1* being the most interconnected mutation, consistent with its known central role in AML pathogenesis[10,31]. For example, the network includes a connection between *NPM1* and *FLT3*.

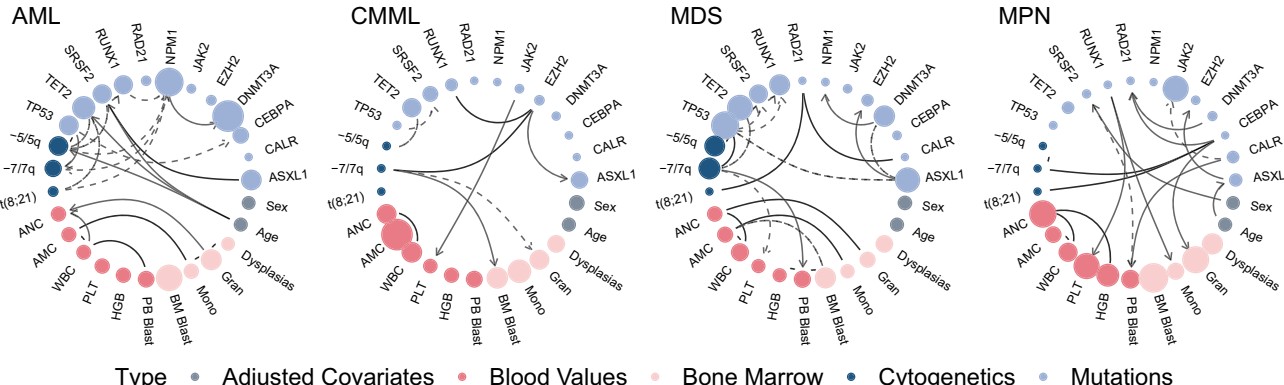

**Fig. 2 | Cancer-type specific networks.** Genomic features are illustrated in blue, blood and bone marrow values in red and demographic variables in grey. The node size for mutations and cytogenetic variables corresponds to their frequency within each cluster, with more frequent variables appearing as larger nodes. In contrast, the node size of continuous variables, like PB blast counts, gets larger when the values of those variables are higher. Directed edges signify the deduced direction of probabilistic (conditional) dependency, while undirected edges denote instances where this could not be determined. Solid edges represent positive correlation while dashed edges represent negative correlation. Displayed are the most frequent and interconnected variables; the full networks can be found in Supplementary Fig. S1. AML acute myeloid leukemia, CMML chronic myelomonocytic leukemia, MDS myelodysplastic syndromes, MPN myeloproliferative neoplasms. AMC absolute monocyte count, ANC absolute neutrophil count, BM bone marrow, HGB hemoglobin, PB peripheral blood, PLT platelet, WBC white blood count. Source data are provided as a Source Data file.

*NPM1* mutations are often accompanied by *FLT3* mutations, which can result in a poorer prognosis than *NPM1* mutations alone[32,33]. Chromosomal abnormalities in chromosome 5 (-5/del(5q)) were most frequent and also notably connected with other chromosomal abnormalities and *TP53* mutations, underlying the importance of this gene for chromosomal stability[34]. Our network displays the interaction between inv(16) and *KIT* mutations (Supplementary Fig. S1). The presence of *KIT* mutations in patients with inv(16) has been identified as a factor contributing to an increased risk of relapse and adverse outcomes compared to those without *KIT* mutations[35]. We observe that age is highly connected to the mutations *SF3B1*, *SRSF2*, and *TET2*, and the chromosomal abnormality -5/del(5q), which may reflect leukemias deriving from previous MDS clones, resulting as an accumulation of genetic driver events later with age.

In the MDS network, the most frequent mutations were *TP53*, *TET2*, *RUNX1*, *DNMT3A*, and *ASXL1*, with typical myelodysplasia-related chromosomal abnormalities -5/del(5q) and -7/del(7q) appearing frequently. Our network elucidates interconnections of -7/del(7q) with various genomic and clinical features, corroborating its relevance in MDS classification and prognosis. The *ASXL1* mutation is highly connected to various other mutations in our network and has been previously found to be associated with adverse outcomes in MDS patients[36]. Notably, our network displays interconnections among *ASXL1*, *SRSF2*, and *RUNX1*. This combination has been shown to be synergistic in contributing to poor prognosis in MDS[37].

In the CMML network, the most frequent mutations included *TET2*, *SRSF2*, and *ASXL1*. −7/del(7q) was found to be highly interconnected, showing an effect on bone marrow blast and bone marrow granulocytes, which supports previous observations of its association with disease progression and poor prognosis[38]. The network displays interconnections among *TET2* mutations and -5/del(5q). *TET2* mutations were significantly associated with the presence of unfavourable cytogenetic abnormalities, including -5/del(5q)[19].

Lastly, in the MPN network, the most frequent mutation was not surprisingly *JAK2*. Our network displays the interaction among the *JAK2*, *CALR*, and *MPL* mutations, which are known for their mutual exclusivity[39] and are associated with different MPN subtypes. *CEBPA* was highly interconnected to various chromosomal abnormalities and mutations, indicating its potential importance in the complex genomic landscape of MPN patients[40]. In addition, our network displays the co-

occurrence of the *JAK2* mutation and del(20q), a chromosomal abnormality associated with more aggressive disease phenotypes[41].

## Pan-Myeloid Leukemia Subgroups

As a next step, we clustered the cohort of 1323 patients diagnosed with AML, MDS, MPN, and CMML together based on their genomic and clinical features. Through this approach, we identified pan-myeloid leukemia subgroups that cut across traditional cancer classifications.

Figure 3a displays a two-dimensional projection of the patient samples coloured by cancer type, reflecting how well they fit to the learned cluster-specific networks (see also Supplementary Fig. S2b). The projection employed (multi-dimensional scaling) preserves distances as well as possible given the need to reduce to two dimensions, so closer points are generally more similar patient profiles, though the mapping is not exact. Patients diagnosed with MDS predominantly cluster on the left side of the plot, while those diagnosed with AML are primarily situated on the right. CMML patients largely localize to the lower region of the plot. Notably, patients with MPN manifest as a distinct group, occupying the bottom-right corner of the density plot. By clustering all patients in an unsupervised manner, we cut across their traditional classifications as illustrated in Fig. 3b (and Supplementary Fig. S2a). In particular, we identified several clusters that lie at the intersection of AML, MDS, and CMML. Though we condition on age in the clustering assignment, since it may influence other variables and their connections, different age distributions across clusters may be expected, reflecting their different compositions in terms of cancer types as well as mutations. The distributions largely overlap since the conditioning ensures that the clustering is not driven by age itself (Supplementary Fig. S3).

As shown in Fig. 3c, the nine learned subgroups can be summarized into three mixed AML-MDS subgroups (clusters B, C, D), two MDS-specific subgroups (clusters A, G), two AML-specific subgroups (clusters H, E), one CMML-MDS subgroup (cluster I) and an MPN-specific subgroup (cluster F).

These subgroups are highly significant predictors of survival, even after accounting for both clinical and histopathological variables such as age, sex, and cancer type in the survival analysis (likelihood ratio = 106.8; $p = 9.1 \times 10^{-42}$). A comprehensive summary of the survival analysis, including median overall survival times stratified by cancer type, is provided in Supplementary Table S3 and the corresponding

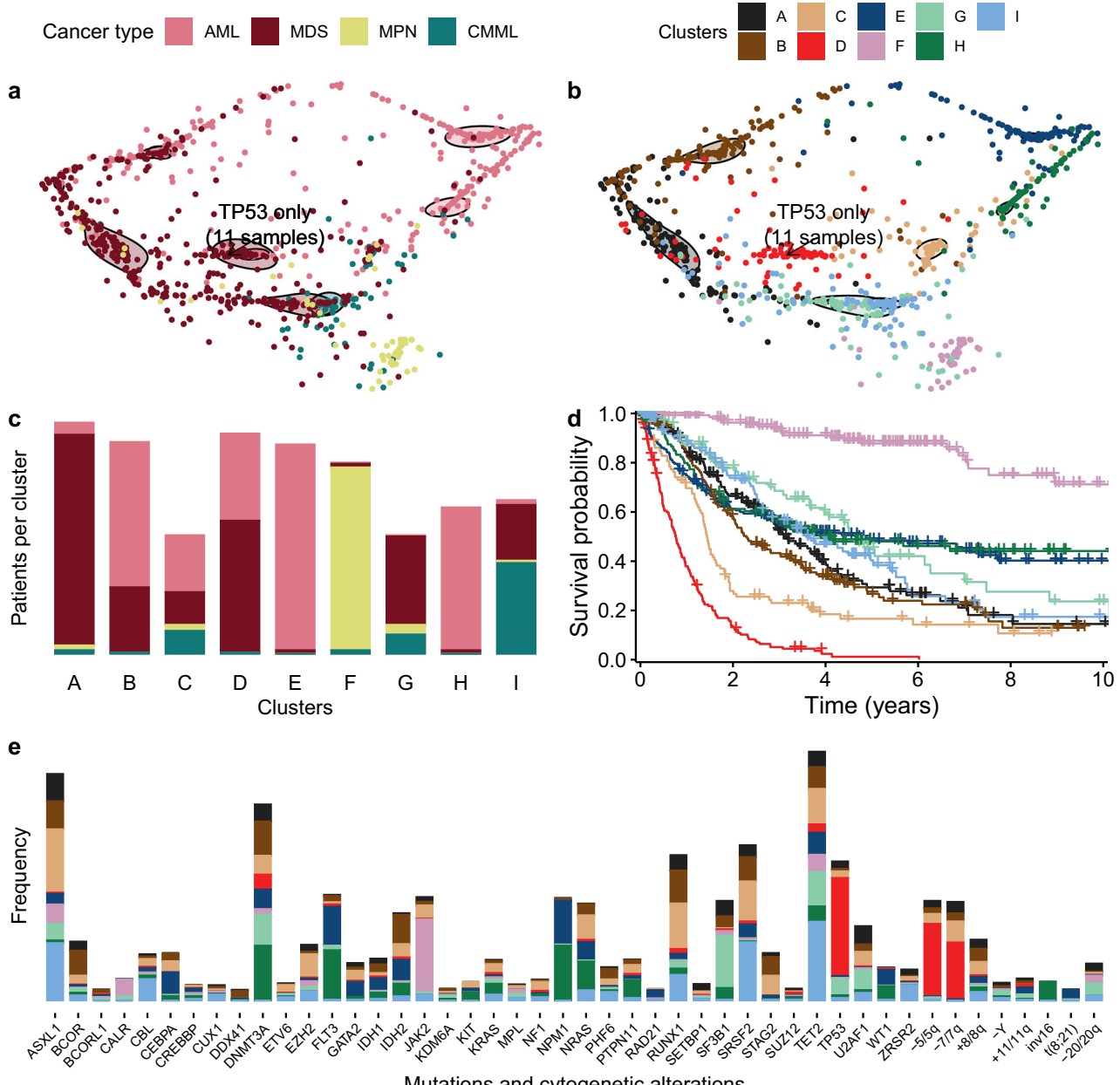

**Fig. 3 | De novo pan-myeloid subgroups. a, b** Patient samples across myeloid malignancies are visualised in a 2D multidimensional scaling projection based on how well they fit to each cluster-specific network. Each dot represents a patient and is coloured by cancer type (**a**) or cluster membership (**b**). Solid-line-encircled shapes depict contours that collectively encompass 50% of the corresponding cluster. Enlarged versions can be found in Supplementary Fig. S2. **c** Distribution of the patient samples across the clusters. **d** Kaplan-Meier survival curves illustrate the survival probabilities associated with each cluster. **e** Barplot displaying the frequency of mutations and cytogenetic alterations across the nine clusters. AML acute myeloid leukemia, CMML chronic myelomonocytic leukemia, MDS myelodysplastic syndromes, MPN myeloproliferative neoplasms. Source data are provided as a Source Data file.

Kaplan-Meier survival curves for each cancer type are provided in Supplementary Fig. S4.

Though the clustering is a discretisation of the each patient's assigned weight to the different clusters, the continuous weights (reflected in the positioning in the two-dimensional projection, Fig. 3a, b) may be informative too. For example, if we stratify patients in cluster D by their similarity to cluster A, we find a significant difference (Supplementary Fig. S5).

With the clustering we can see the importance of including and appropriately adjusting for the clinical covariates. Without adjustment our survival predictions are worse, while they worsen further when considering mutations only (Supplementary Table S4). Likewise, using prior knowledge from the STRING database (Methods) aids the network learning as excluding this also leads to worse survival prediction (Supplementary Table S4 last row) with changes in cluster assignment shown in Supplementary Fig. S6. We can additionally see the benefit of clustering in a pan-myeloid setting. For example, if we cluster the AML and MDS patients together we obtain better predictions than treating the two separately (Supplementary Table S5).

Further, we developed an interactive web portal to facilitate the application of our findings to other patient samples for research purposes. Available at https://myeloid-prediction.ethz.ch, the portal allows users to classify patient samples into our pan-myeloid subgroups.

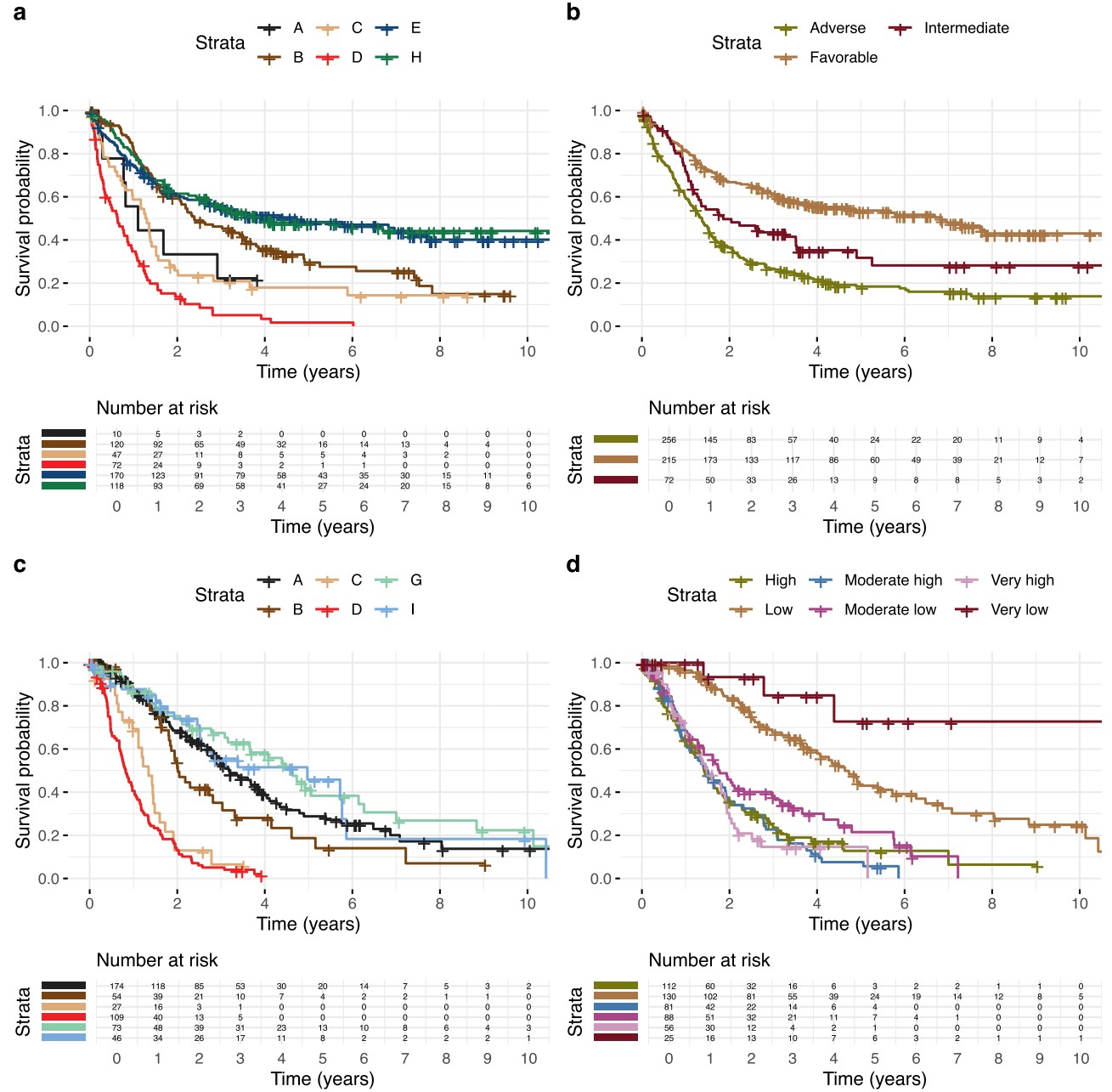

**Fig. 4 | Kaplan-Meier survival curves comparing distinct clusters and ELN2022 and IPSS-M risk classifications in AML (acute myeloid leukemia) and MDS (myelodysplastic syndromes), respectively.** Displayed are the survival probabilities for distinct clusters in AML patients (**a**) and their survival outcomes stratified by ELN2022 risk classification (**b**). Corresponding analyses for MDS with survival probabilities across distinct clusters (**c**) and according to IPSS-M risk classification (**d**). Groups with fewer than 10 individuals are omitted for clarity.

**Comparison to Risk Scores.** We compared our cancer subgroups with established risk stratification models, specifically the European Leukemia Net 2022 (ELN2022) for AML patients[9] and the International Prognostic Scoring System-Molecular (IPSS-M) for MDS[17]. We note that although our cancer subgroups are based on the genomic, clinical and demographic data of the patients, they are not informed by survival data, making this an independent validation, while the established risk scores are based on survival data and target predicting outcome directly. After adjusting for clinical and histopathological covariates, our survival analysis showed that while ELN2022 serves as a significant predictor for survival in AML (likelihood ratio $= 16.9$; $p = 4.2 \times 10^{-5}$), our subgroups offer superior predictive power (likelihood ratio $= 29.2$; $p = 9.0 \times 10^{-10}$). A similar enhancement is observed for MDS when comparing IPSS-M (likelihood ratio $= 53.9$; $p = 1.1 \times 10^{-19}$) to our

subgroups (likelihood ratio $= 76.4$; $p = 5.2 \times 10^{-29}$). Notably, combining our subgroups with existing risk scores further elevates the predictive accuracy for survival both in AML and MDS (Supplementary Table S6). Figure 4 shows the corresponding Kaplan-Meier survival curves of our subgroups and those derived from existing risk scores. A summary of the survival analysis and median overall survival times for the various risk scores can be found in Supplementary Table S7, while the assignment to clusters and risk scores is depicted in Supplementary Fig. S7.

### AML-MDS Clusters
**Cluster D: Very High-Risk AML-MDS Subgroup.** Cluster D emerged as the subgroup with the worst survival prognosis across all identified clusters. Notably, 40% of patients in this cluster had AML, and

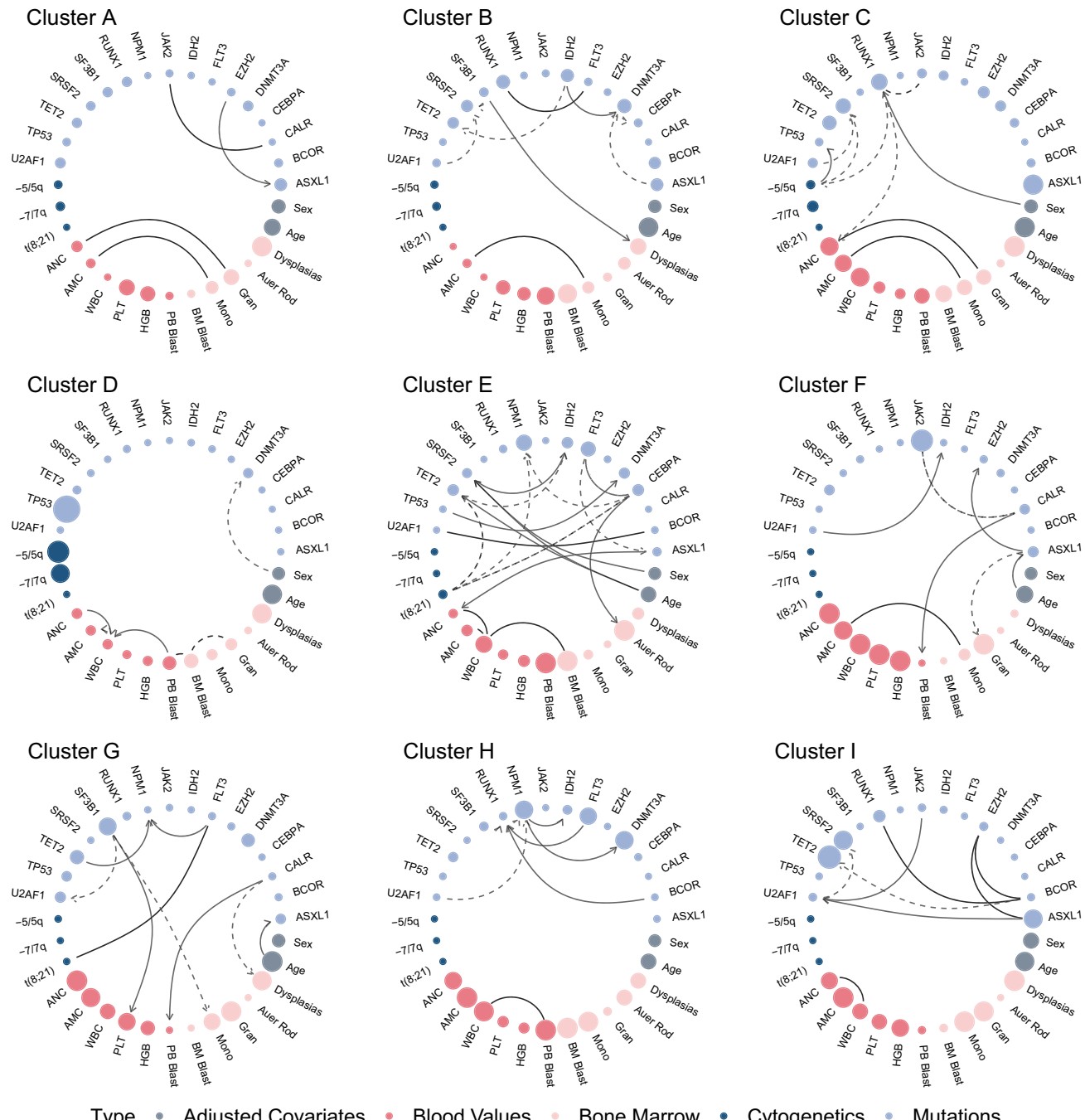

**Fig. 5 | Cluster-specific networks of the clusters A-I.** Genomic features are illustrated in blue, blood and bone marrow values in red and demographic variables in grey. The node size for mutations and cytogenetic variables corresponds to their frequency within each cluster, with more frequently mutated genes appearing as larger nodes. In contrast, the node size of continuous variables gets larger when the values of those variables are higher on average. Directed edges signify the deduced direction of dependency, while undirected edges denote instances where the dependency direction could not be determined. Solid edges represent positive correlation while dashed edges represent negative correlation. Displayed are the most frequent and most interconnected variables; the full networks can be found in the Supplementary Fig. S8. AMC absolute monocyte count, ANC absolute neutrophil count, BM bone marrow, HGB hemoglobin, PB peripheral blood, PLT platelet, WBC white blood count. Source data are provided as a Source Data file.

60% had MDS. The median overall survival (OS) for this cluster was only 0.8 years (0.6 years for the AML patients and 0.8 years for the MDS patients). These figures are considerably worse than the highest risk groups defined by the IPSS-M and ELN2022 classifications, which exhibit median OS of 1.4 and 1.5 years in our study cohort, respectively, even with the same number of groups across the classifications of AML and MDS. As further shown in Fig. 3b, the patients in cluster D localise at the intersection of AML and MDS

diagnoses. This cluster exhibited the highest frequency of *TP53* mutations, along with frequent chromosomal alterations -5/5q and -7/7q (Fig. 5). A striking 90.1% of patients in this cluster harbored *TP53* mutations, emphasizing its potential role in driving poor outcomes. However, some patients with *TP53* mutations were assigned into lower-risk clusters, specifically those lacking cytogenetic alterations. Intriguingly, this cluster displayed minimal mutations in other significant prognostic genes, such as *FLT3*,

*ASXL1*, and *RUNX1*, reinforcing the unique genetic makeup that defines this high-risk subgroup.

**Cluster C: High-Risk Subgroup.** Cluster C, second only to Cluster D in its poor survival prognosis, had a median OS of 1.4 years. Located at the intersection of AML, MDS, and CMML, this cluster was composed of 48% AML, 28% MDS patients, and 19% CMML patients. Genomic features that were prominently represented included *RUNX1*, *ASXL1*, and *SRSF2* mutations. Additionally, our network analysis revealed high interconnectivity between *RUNX1*, *SRSF2*, and -5/5q (Fig. 5), suggesting potential relationships among these mutations and chromosomal alterations.

**Cluster B: Moderate-Risk Subgroup.** With moderate risk of death, Cluster B exhibits the third worst prognosis among the groups, with a median OS of 2.3 years. The cluster comprises 69% AML and 31% MDS patients. This cluster is characterized by a high frequency of *RUNX1* and *DNMT3A* mutations. Particularly noteworthy is the heightened interconnectivity of *RUNX1* within this cluster (Fig. 5), implying its potential role as a key driver in this subgroup. In addition to these primary mutations, Cluster B presents with very high blast levels, as well as with many other moderate-frequency mutations and cytogenetic alterations.

### MDS-Specific Clusters

**Cluster A: Moderate-Low Risk Subgroup.** Comprised primarily of MDS patients (91%), Cluster A exhibits a moderate-low risk profile with a median OS of 3.2 years. Genomic features frequently observed were *DNMT3A* and *ASXL1* mutations, and network analysis in Fig. 5 revealed significant interconnections between *ASXL1* and other chromatin modifiers like *SUZ12* and *EZH2*.

**Cluster G: Low-Risk Subgroup.** Cluster G mainly included MDS patients (74%) and presented the best survival prognosis among the MDS-specific clusters, with a median OS of 4.6 years. This cluster was characterized by a high prevalence of mutations in *TET2*, *SF3B1*, and *DNMT3A*.

### AML-Specific Clusters

**Cluster H: Moderate-Low Risk Subgroup.** Comprising almost exclusively AML patients (98%), Cluster H displayed a moderate-low risk profile, with a median OS of 3.9 years. Notably, all patients with inv(16) mutations were grouped in this cluster, and the gene most interconnected was *NPM1*, which had 6 interconnections within the cluster (Fig. 5).

**Cluster E: Low-Risk Subgroup.** Cluster E, another predominantly AML cluster (98%), had the best prognosis among the AML-specific clusters with a median OS of 4.7 years. Genomic features that were commonly observed included mutations in *FLT3*, *NPM1*, and particularly *CEBPA*, which showed the highest level of interconnectivity within the cluster (Fig. 5).

### CMML-MDS and MPN-Specific Clusters

**Cluster I: Moderate-Low Risk Subgroup.** Comprising 59% CMML and 36% MDS patients, Cluster I presented a median OS of 3.6 years. High frequencies of *TET2*, *SRSF2*, and *ASXL1* mutations were observed, and network analysis in Fig. 5 indicated that these features were highly interconnected.

**Cluster F: Low-Risk MPN Subgroup.** Predominantly comprised of MPN patients (96%), this cluster exhibited the best overall survival across all clusters. The genomic landscape was dominated by *JAK2* and *CALR* mutations, which were also interconnected in the cluster-specific network (Fig. 5). Notably, patients outside this predominantly MPN subgroup face significantly worse survival outcomes.

### Classifying Unseen Cohorts with our Subgroups

With the inferred clusters from our pan-myeloid cohort, and the network-based model describing each cluster, we can classify additional patients into our nine subgroups. To see how well our subgroups generalise we analysed and classified an AML cohort of 3626 patients[15] and an MDS cohort of 875 patients[14]. The Kaplan-Meier plots stratified by subgroup assignment of risk score classifications (Supplementary Fig. S9) are similar to those for our cohort (Fig. 4). Again we see a strong advantage in using our subgroup assignment compared to the ELN2022 classification for the validation AML cohort, though only a moderate improvement compared to the IPSS-M classification for the validation MDS cohort (Supplementary Table S8). However, when we include our subgroup assignments in addition to the risk score classifications, we still see a significant improvement in predicting survival (Supplementary Table S8, last two rows).

### Numerical Simulations

To evaluate the performance of CANclust, our covariate-adjusted network clustering approach (Methods and Supplementary Section A), we benchmarked it against conventional clustering methods. Various simulations were performed to emulate typical pan-cancer scenarios, including cluster-specific probabilistic relationships and causal effects from covariates on clustered variables. The chosen conventional clustering algorithms for benchmarking include network-based methods, such as the Bayesian Network Mixture Model (BNMM), and traditional, non-graph-based methods like k-means and the Bernoulli Mixture Model (BMM).

In the presence of clinical covariates, our covariate-adjusted method achieved the highest clustering accuracy (Supplementary Section B), aligning well with our theoretical expectations (Supplementary Section A.1). Generally, network-based clustering methods exhibit higher clustering accuracy, reflecting their ability to model probabilistic relationships among variables. It is noteworthy that the inclusion of additional covariates in traditional, non-graph-based models led to a decrease in performance. This decline can be attributed to these models' assumption of independent probability distributions, reflecting their limitation in capturing covariate-influenced interactions. A comprehensive evaluation, including benchmarks for varying cluster-specific parameters, can be found in Supplementary Section B.

### Method Validation on TCGA Data

To further validate the performance of our method, we applied it to a large-scale pan-cancer dataset from TCGA[30], which includes the mutational profiles and clinical information of 8085 patients from 22 different cancer types.

Leveraging our covariate-adjusted approach to integrate clinical covariates and genetic features, we identified cluster assignments with predictive value for survival outcomes. This predictive ability holds even after accounting for clinical and histopathological covariates in the survival analysis (likelihood ratio = 46.6; $p = 1.0 \times 10^{-10}$). Further details are presented in Supplementary Section C.

Importantly, our cluster assignments outperformed those from traditional clustering methods that do not account for clinical covariate effects, achieving a higher likelihood ratio in the Cox proportional hazards regression model (see Supplementary Table S9). The elevated likelihood ratio suggests that the covariate-adjusted clusters are more accurate in predicting survival, thus confirming both our theoretical expectations and the results from numerical simulations.

## Discussion

We introduced a network-based patient clustering method that integrates genomic and clinical data based on their distinct probabilistic relationships. This method not only demonstrated superior clustering performance against conventional algorithms in simulations, but also showed its effectiveness on a large-scale pan-cancer dataset encompassing 8085 patients. Critically, integrating genomic and clinical data significantly improved the survival predictions compared to using mutational profiles alone. This underscores the enhanced predictive power gained through the integration of genomic and clinical data and their interactive relationships, offering a more comprehensive understanding of tumor behavior and progression. While there are many clustering algorithms available, the graphical nature of our approach offers a transparent lens to decipher the complex interplay of the genetic and clinical variables.

Through this approach, in a cohort of 1323 patients, we identified nine subgroups across myeloid malignancies that cut across traditional classifications of AML, MDS, CMML and MPN. These subgroups were highly significant predictors of survival, even after accounting for both clinical and histopathological variables such as age, sex, and cancer type. For each subgroup, we identified subgroup-specific patterns among mutations, cytogenetic alterations, and clinical covariates. These patterns, represented by networks, elucidate interactions among the variables that may aid identifying therapeutic targets and strategies. Learning de novo cancer subgroups independent of the cancer type, we find that traditional classifications of myeloid malignancies are partially preserved in these subgroups. For instance, nearly all MPN patients are assigned to one cluster, reinforcing the distinct genomic and clinical landscape of MPN relative to other myeloid malignancies. Notably, MPN patients outside the predominant MPN cluster exhibit a trend toward worse survival rates, hinting at potential transformations in their disease status.

The emergence of high-risk mixed clusters comprising both AML and MDS patients unveils overlaps in their underlying genomic and clinical landscapes. The particularly poor prognosis observed for MDS patients within these mixed clusters may suggest an impending transformation into AML. Since MDS is a heterogeneous disease, this aligns with the general observation that higher-risk MDS behaves more similarly to AML including for *TP53*-altered subtypes[42]. Historically, higher-risk MDS has also been treated with AML-like regimens[43,44], and ongoing clinical trials are exploring the use of AML-approved treatments in higher-risk MDS (VERONA trial: NCT04401748). Our clustering might then help predict the subset of patients who might benefit from AML-like therapies. Notably, within these high-risk, mixed subgroups, AML patients fare worse than their counterparts in AML-dominant subgroups, potentially indicating a history of disease transformation to secondary AML.

Our results support the growing trend to consider AML and MDS as a spectrum rather than distinct entities[42], underscoring the considerable overlap in their genomic and clinical features. Clustering AML and MDS patients together in a unified cohort allowed us to unveil subgroups that lie at the intersection of both cancer types and are characterised by exceptionally poor survival prognosis. This underscores the imperative to jointly analyze these cancer types when deriving subgroups or formulating risk scores in future studies.

We compared our cancer subgroups to established risk scores, specifically ELN2022 for AML and IPSS-M for MPN patients. Our subgroups manifested a significant enhancement in predictive accuracy with regard to survival outcomes for both AML and MDS cohorts. Moreover, using our cancer subgroups in addition to the risk scores allowed for much more accurate survival predictions than using the risk scores alone. Unlike the risk scores which focus on a few low-dimensional features predictive of outcome, our network-based approach treats the entirety of the data holistically including its complex interdependencies also with clinical covariates. This allows it

to better model the data, and even predict survival better despite being trained independently without access to the survival data. To enhance the accessibility of our findings for research purposes, we have developed an interactive web portal that enables straightforward classification of patient samples into our cancer subgroups (https://myeloid-prediction.ethz.ch/).

Our method is available as a readily deployable software package (https://CRAN.R-project.org/package=clustNet), so future research can adapt it to explore other cancer types that share overlapping features or have the potential to transform into the same advanced-stage disease. We anticipate that applying this methodology to larger, more diverse cohorts may provide increasingly granular insights, as we observed when clustering AML and MDS together. While our investigations predominantly considered bulk DNA sequencing data, cytogenetic features, and clinical attributes, the leukemia cellular hierarchy has also been shown to associate with genomic and clinical properties[45] and integrating our approach with additional data facets, like such transcriptomic data, could further refine the subgroups and enhance their prognostic value. Handling multi-omic data in network learning has previously been considered[25] and could be combined with our method to integrate clinical covariates. Moreover, different transcriptomic programs can co-exist within the same cancer and be linked to different genetic subclones[46], while single-cell DNA sequencing has further illuminated the wealth of subclones and evolutionary patterns in AML, along with their links to phenotypic heterogeneity[47]. These results suggest that in addition to data integration, treating cancers at the subclonal level could also lead to better modelling of subgroups and prognosis.

Although we incorporate tens to hundreds of features across thousands of patients, the computational cost grows with the number of features. For example the clustering takes minutes for small examples with 20 features, several hours for the pan-myeloid data with around 60 and just over a day for the TGCA data with 200 (see Supplementary Section A.2). The scenario could then become more computationally demanding with additional data layers, or where feature counts can grow into the thousands. In such high-dimensional settings, computational costs could constrain applicability, requiring further optimizations to handle the increased resource demands efficiently. One approach to reducing these demands is to enforce sparsity through regularising the network to focus on the strongest connections. With sufficient regularisation, the network learning can be divided into smaller sub-problems and even reduced to polynomial complexity[48].

In our analysis, we summarised mutations at the gene level, though different point mutations in the same gene may behave differently. Increasing the granularity, and resolving mutations down to individual genomic positions, likewise increases the dimensionality of the network, along with the sparsity of the data and the complexity of network learning. For certain mutations, however, accounting for their potentially different effects in the same gene may still increase predictive accuracy. One way to collate this to summarise at the gene level is to combine the predicted pathogenicity of each mutation, for example with the Ensembl variant effect predictor[49] which includes the SIFT[50] and PolyPhen[51] predictors. More recently, the effects of mutations have been summarised through neural network models trained on cell-line drug-response data[52]. These types of continuous summaries could then be used to replace binarised values in analyses such as ours. Alternatively, mutations could be combined at the pathway level, which could also reduce the dimensionality of the networks.

Overall, our network-based clustering method provides an integrative approach to understanding the genomic and clinical landscape across myeloid malignancies. Through this prism, not only do we achieve improved patient stratification, but we also unearth nuanced patterns and interactions that hold potential to guide stratification for clinical trials and the development of targeted therapeutic strategies.

## Methods

### Pan-Myeloid Malignancy Data

Our research complies with all relevant ethical regulations, and the study protocol was approved by the Institutional Review Board of MD Anderson Cancer Center.

The pan-myeloid dataset encompasses the mutational profiles and clinical covariates of 1323 patients diagnosed with AML, MDS, MPN and CMML. The cohort, sourced from MD Anderson, included 543 patients with AML, 492 with MDS, 118 with CMML, and 170 with MPN. The age distribution within the cohort ranged from 18 to 94 years, with a median age of 66 (Supplementary Table S1). The data comprises mutational profiles, cytogenetic profiles, blood values, bone marrow values and demographic features (Supplementary Table S2). Cytogenetics was performed by global karyotyping of each sample, while mutations were detected through an 81-gene targeted sequencing panel based on recurrent somatic mutations in myeloid malignancies[53], which form part of the clinical sequencing used at MD Anderson for clinical decision making. Mutations were summarised at the gene level and recorded as binary values indicating the presence of any mutation in that gene or their absence. Features with a prevalence below 1% were excluded, narrowing our focus to 38 genes and eight cytogenetic profiles.

We identified variables from both blood and bone marrow data that exhibited significant deviations from the expected normal ranges in a healthy population, resulting in seven key blood metrics and four crucial bone marrow metrics. As illustrated in Supplementary Fig. S10, these selected clinical covariates exhibit cancer-specific trends, implying their potential utility in deriving biologically meaningful cluster assignments within and across myeloid malignancies. For instance, the values for AML, CMML, and MDS patients largely diverged from healthy norms, demonstrating classic clinical manifestations such as anemia and thrombocytopenia. In contrast, MPN samples exhibited elevated levels of platelets, hemoglobin, white blood cells, and absolute neutrophil count, reflecting the proliferative aspects of this malignancy.

We discretized continuous variables into two levels, using the median as a cutoff value to ensure a balanced distribution of samples across these categories. For blast values, we employed specific cutoff points grounded in established hematopathological classifications. Specifically, we categorized bone marrow (BM) blasts into three ranges: zero to 10, 11 to 20, and 21 to 100. Peripheral blood (PB) blasts were grouped into two categories: one consisting of samples with zero blasts and another comprising counts between 1 and 100.

### Pan-Cancer Data

To assess the performance of our method, we further applied our methodology to a dataset from The Cancer Genome Atlas (TCGA)[30], which comprises the mutational profiles and clinical information of 8085 patients across 22 different cancer types (Supplementary Section C). We selected the 16 most frequently mutated genes for each primary cancer type, resulting in an aggregate of 201 genes analyzed across all cancers.

### Covariate-Adjusted Clustering

We want to cluster genomic and clinical profiles of tumors into $K$ different groups. To account for cluster-specific probabilistic relationships, we model each cluster by a Bayesian network. Each Bayesian network $(\mathscr{G}, \theta)$ comprises a directed acyclic graph $\mathscr{G}$ and associated local probability distributions $\theta$. The nodes in the Bayesian networks represent mutations and clinical covariates. We differentiate between two different types of covariates: (1) *cluster-independent covariates* that can only have outgoing edges into the mutations, and (2) *cluster-dependent covariates* that may have incoming edges from the mutations.

Exploiting our knowledge about the covariates, we can correspondingly make assumptions about the direction of the causal pathways. As an example, the covariate sex may make specific mutations more likely to occur, but it is not the consequence of specific mutations. Thus, the covariate sex has a downstream causal effect on the mutations and is therefore a cluster-independent covariate. In contrast, in myeloid malignancies, the covariate cancer type is determined based on the mutational profile of the patients[11], making it a cluster-dependent covariate.

Since we are interested in mutational patterns, we do not want to cluster patients based on cluster-independent covariates such as age and sex. For instance, if clustering were conducted based on age, the resulting groupings would primarily feature older patients in one set and younger ones in another. This approach would only echo an already known covariate, thus offering little new insight into the fundamental mutational patterns. Instead, we are interested in finding additional insights beyond the cluster-independent variables. Learning subgroups that cut across the cluster-independent variables may allow to further sub-stratify some of the more heterogeneous groups that are defined by the cluster-independent variables. We therefore need to correct for the effect of cluster-independent covariates on the clustering. In contrast, cluster-dependent covariates can carry information about the mutational patterns. Hence, they can be modelled analogously to the mutations without further adjustment. With this setup, we assume the generating probability distribution of the cluster-dependent variables $X_V$ to be conditioned on the cluster-independent covariates $X_C$:

$$p(X_V) = \sum_{k=1}^{K} \pi_k P(X_V | X_C, \mathscr{G}_k, \theta_k) \tag{1}$$

where $\sum_{k=1}^{K} \pi_k = 1$ is the weight of each cluster, and the probability of $X_V$ given cluster $k$ is

$$P(X_V | X_C, \mathscr{G}_k, \theta_k) = \prod_{i \in V} P\left(X_i | X_C, X_{pa(i)_k}, \theta_k\right) \tag{2}$$

Knowledge of the generating distribution allows us to define a mixture model to cluster the mutational profiles. We learn the membership probabilities $\tilde{\phi}(X_V | k)$ of the mutations for each cluster using the EM-algorithm similar to[24,54]. Extending the previous approach, we adjust for the effects of clinical covariates on the clustering by employing the following adjusted membership probability function

$$\tilde{\phi}(X_V | k) = \frac{\chi + \gamma_k \cdot P\left(X_V | X_C, \hat{\mathscr{G}}_k, \hat{\theta}_k\right)}{\chi K + \sum_{k'=1}^{K} \gamma_{k'} \cdot P\left(X_V | X_C, \hat{\mathscr{G}}_{k'}, \hat{\theta}_{k'}\right)} \tag{3}$$

where $\gamma_k = \frac{\sum_{1=1}^{N} \tilde{\phi}(X_V | k)}{N}$ is the weight of each cluster and $(\hat{\mathscr{G}}_k, \hat{\theta}_k)$ are the learned Bayesian network parameters. In this adjusted function, the probability of the mutations is conditioned on the covariates. This has two major advantages: first, it corrects for confounding that might be induced when neglecting the covariates, and second, it adjusts for the effects the covariates might have on the clustering. We prove that this correction leads to a lower variance in the membership probability function in Supplementary Section A.1 and outline the individual steps of our algorithm in Supplementary Section A.2.

Calculating $P(X_V | X_C, \hat{\mathscr{G}}_k, \hat{\theta}_k)$ is a marginalization problem, which is NP-hard in general for Bayesian networks[55] and typically requires approximations in high-dimensional settings[56]. However, $P(X_V | X_C, \hat{\mathscr{G}}_k, \hat{\theta}_k)$ simplifies under the condition that the covariates have only outgoing edges to the mutations due to the factorization property of Bayesian networks[57]. This significantly reduces the computational cost and allows for an exact calculation of the membership probability for each sample.

## Network learning

The networks were learned via Bayesian structure learning[58,59], that was informed by prior knowledge about the expected edges. Known human functional interactions among the 38 selected genes were imported from the STRING database[60]. When learning the Bayesian network structure, edges between two interacting genes from the STRING database were not penalized, while all other edges were penalized by a factor of two.

We restricted the search space of the edges by applying knowledge about the expected directionality of the edges across different data modalities. Specifically, edges going from the blood and bone marrow values into the genomic variables were not allowed in our search space, reflecting the likely causal direction. Cluster-independent variables only had outgoing edges into all other variables.

## Robustness

Clustering high-dimensional data involves exploring the large space of possible patient groupings, which may be challenging and contain many local optima. We therefore wished to check the robustness of our clustering procedure. As assigning an equal weight to each patient is an unstable equilibrium clustering solution, small perturbations can lead to the clustering algorithm initially exploring diverse parts of the clustering space. We then check the robustness of the final cluster assignments.

In particular, we ran the entire clustering algorithm 100 times from different starting perturbations. For the starting point, from an equal weight per patient, we increased the relative weight by 0.1% independently for each patient to a uniformly sampled cluster. As the measure of robustness we computed the adjusted Rand index (ARI) of the different clustering outcomes compared to the clustering reported in the main text.

Overall the results are highly stable (Supplementary Fig. S11) with a typical ARI of around 0.92, though there is a small amount of stochastic variation that can arise in clustering such high-dimensional data.

## Network Visualization

The visual attributes of the network nodes are indicative of mutation frequency, cytogenetic alteration rates, and the magnitudes of clinical covariates. Specifically, for binary mutational and cytogenetic variables, the node size is proportionate to their prevalence within each cluster; thus, genes with a higher mutational incidence manifest as enlarged nodes. The node size of continuous variables gets larger when the values of those variables are higher; for instance, elevated PB blast counts are reflected by a larger node size for the PB blast variable. For categorical covariates, such as sex, the node size is determined by a normalized cluster-specific entropy measure, serving as a visual metric of the variable's informativeness for that cluster. To illustrate, if the number of female individuals within a cluster is higher than in the population average, the corresponding node will appear larger. We define a cluster-specific entropy measure as the normalized difference between the entropy across the clusters and the cluster-specific entropy. The overall Shannon entropy across all clusters for a variable $X$ with categories $\mathscr{X}$ is defined as

$$H(X) = -\sum_{x \in \mathscr{X}} p(x)\log(p(x)), \tag{4}$$

where $p(x)$ is defined by the relative frequency of $x \in \mathscr{X}$ across all clusters. Analogous, the cluster-specific Shannon entropy is defined according to the relative frequency of $x \in \mathscr{X}$ across cluster $k$

$$H_k(X) = H(X)|_{p(x) = p(x|k)} = -\sum_{x \in \mathscr{X}} p_k(x)\log(p_k(x)). \tag{5}$$

The normalized cluster-specific entropy measure $S_k(X)$ of variable $X$ reflects the relative entropy, which we define as the difference in cluster-specific and overall entropy

$$H_k^{\Delta}(X) = H_k(X) - H(X) \tag{6}$$

on which we applied the following normalization

$$S_k(X) = 1 - \frac{H_k^{\Delta}(X) - |\min(H_k^{\Delta}(X)|}{|\max(H_k^{\Delta}(X)| + |\min(H_k^{\Delta}(X)|}. \tag{7}$$

Given the large number of mutations and covariates, Fig. 2 and Fig. 5 display only the most frequent and most interconnected variables; the full networks can be found in Supplementary Fig. S1 and Supplementary Fig. S8, respectively.

## Survival analysis

We applied the Cox proportional hazards regression model, accounting for the strong predictors of survival: age, sex, and cancer type. By controlling for these factors, we isolated the influence of cluster assignments on survival outcomes. Further, we juxtaposed our cancer subgroups with the established risk stratification models ELN2022 for AML[9] and IPSS-M for MDS[17] (Supplementary Table S6). Survival differences across clusters were visualized using Kaplan-Meier curves, offering a graphical representation of survival probabilities over time in relation to the cluster assignments. The Kaplan-Meier survival plots for each cancer type are shown in Supplementary Fig. S4 and a side-by-side comparison of our subgroups with the risk scores ELN2022 and IPSS-M are shown in Fig. 4.

## Reporting summary

Further information on research design is available in the Nature Portfolio Reporting Summary linked to this article.

## Data availability

Our GitHub repository at https://github.com/cbg-ethz/myeloid-clustering includes a pre-processed version of the publicly available data sourced from TCGA, featuring genomic and clinical data for 8085 patients across 22 different cancer types. The mutational profiles for our patient cohort comprising 1323 individuals diagnosed with myeloid malignancies are also available at the same GitHub repository. The mutations are obtained from clinical sequencing for which the raw sequencing data is controlled under privacy regulations and is not accessible by the investigators. The clinical covariates for each patient are available to all researchers under restricted access for data privacy reasons. Please email to KT (KTakahashi@mdanderson.org) for the request. Source data are provided with this paper.

## Code availability

The methodology developed in this study has been implemented in an **R** package, which is publicly accessible via the Comprehensive R Archive Network (CRAN) at https://CRAN.R-project.org/package=clustNet. This package facilitates network-based clustering of genomic and clinical data, incorporating covariate adjustments. Additionally, an interactive web tool for subgrouping and visualising patient samples can be accessed at https://myeloid-prediction.ethz.ch. All scripts to reproduce the analysis and figures presented in this paper are openly available in a GitHub repository at https://github.com/cbg-ethz/myeloid-clustering.

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

## Acknowledgements

The authors are grateful to acknowledge funding support for this work from the two Cantons of Basel through project grant PMB-02-18 granted by the ETH Zurich (to JK).

## Author contributions

J.K. conceived the project with G.M. F.B. developed the methodology with input from J.K. and G.M. F.B. created the software and performed the formal analyses. M.R., K.M. and K.T. curated the data, with further investigation by M.R. N.B. and J.K. supervised the project. F.B. prepared the original draft with input from M.R., G.M., N.B. and J.K. All authors reviewed and edited the manuscript.

## Funding

## Competing interests

The authors declare no competing interests.
