## [Transparent Peer Review file · Nature Communications]

Network-based clustering unveils interconnected landscapes of genomic and clinical features across myeloid malignancies

Corresponding Author: Dr Jack Kuipers

Version 0:

Reviewer comments:

Reviewer #1

(Remarks to the Author)

Bayer et al introduce a new approach for network-based clustering of cancer patients into groups by mutational and other covariates. They apply their approach to a cohort of 1323 patients diagnosed with various myeloid disorders.

I cannot comment on the mathematics underlying the clustering approach, or its novelty in the machine learning community. My comments take the perspective of an expert on genomics and bioinformatics applications in hematology research.

Superficially, this paper seems a bit like 'yet another clustering of leukemia patients by mutations'. However, it appears that the new clustering algorithm is well suited for scenarios where both genomic and other covariates are present, and thereby goes beyond the state of the art. In particular, this might have exciting applications beyond the inclusion of simple blood count data, see my 3rd point for a suggestion. It would be good to include such an applications to make this paper interesting to a broader readership.

1. For AML and MPN, the clustering obtained outperforms established risk scores (ELN2022 and IPSS-M) (Figure 5). This might not be a fair comparison, since the risk scores were developed on different cohorts, and still perform well on the 1323 patient cohort analyzed here. By contrast, the clusters were obtained on this specific cohort. To make the point that the clusters can be useful in practice, they would need to do this comparison on an independent cohort.

2. The direction of some edges in Figure 2 and 4 creates doubts. For example, DNMT3A mutations generally occur first in leukemia evolution (during clonal hematopoiesis) and is often followed by NPM1, yet the arrow points from NPM1 to DNMT3A. Likewise FLT3 is usually downstream of NPM1.

3. It could be interesting to compare the clusters obtained here to the cell type based classification presented in Zeng et al (10.1038/s41591-022-01819-x). This paper classified AMLs into a spectrum of "Primitive-to-Mature" and "Primitive-to-GMP" based on RNA-seq.

On one side this comparison might shed light on the biological differences between some of the clusters in Figure 3 (e.g. the difference between the NPM1 clusters E and H are not really clear).

On the other hand, the progenitor-based covariates presented in that paper carry higher prognostic value, compared to the dependent covariates currently employed by the authors, and including them into the clustering could further improve the performance here.

In general, RNA-seq based classification of AMLs are becoming increasingly important. The method presented here has the potential for an integrated analysis of mutational data and RNA-seq based features, so including these might take this paper beyond 'yet another clustering of mutations in AML'. If no RNA-seq data is available for the 1323 patients used here, the dataset by Zeng et al could be used.

4. The documentation of the accompanying R package needs to be improved, e.g. the vignette was empty when I tried to open it

5. Figure 2&4: Abbreviations are not defined and the directionality of the arrows is hard to see.

6. Some statements in the discussion (e.g. use of AML therapies in MDS) should be commented on by a reviewer with a clinical expertise, or removed.

Reviewer #2

(Remarks to the Author)

Bayer et al present a data-driven network-based clustering approach that integrates mutational features and clinical covariates to discover de novo cancer subgroups. They apply their approach to a cohort of 1323 patients to identify novel subgroups. Based on their results, this method outperforms established risk classifications in prognostic accuracy. They also apply this method to TCGA data to show differences in the survivals of patient groups. The manuscript raises an important topic however, I have stills some major comments that need to be addressed.

- The clarity of Figure 1 is limited, as it lacks self-description. While I recognize its abstract nature intended to convey the concept, the right panel, particularly the cluster labels, is challenging to decipher. A comprehensive revision and elaboration of the figure legend are essential to enhance understanding.
- While the authors assert in the text that 'To avoid learning the clusters based on demographic covariates such as age and sex, we remove their effect on the cluster assignment by exploiting causal relationships among the variables,' notably, these covariates are still present in the networks depicted in Figure 2. Is it necessary to have them in the networks? If yes, why?
- If the authors were to apply a naive method, clustering tumors based solely on mutational profiles and clinical covariates, without considering causal relationships through a standard clustering method, would the resulting clusters significantly overlap or not with theirs? This analysis could provide valuable insights into the distinct contributions of their proposed method to patient stratification.
- In the cancer-specific and cluster networks, edges solely represent the direction of causal relationships. Nevertheless, in the text, the authors also refer to the mutual exclusivity and co-occurrence of genomic alterations, such as JAK2, CALR, and MPL mutations, as well as JAK2 and del(20q) in MPN. Do they have this specific information? If yes, this can be represented in the networks with different colors of edges.
- In 2D projection for visualization purposes in Figure 3a, which algorithm has been selected? If that technique were changed, still the same representation be obtained?
- Each cluster possesses unique properties and significant pairs within their networks. Maintaining a consistent layout and restricting nodes to the most frequent and highly interconnected variables across all clusters results in sparse cluster-specific networks. I recommend visualizing the clusters by limiting their edges rather than nodes. This approach ensures that the most crucial interactions are highlighted in the main text, and those causal relations specific to clusters are shown in the main figure.
- Regarding the TCGA analysis, they utilized pan-cancer data, including solid tumors. Although the method is originally designed for blood cancers, the rationale behind extending it to pan-cancer data needs further clarification.
- Within each cluster of tumors, the authors reduce the data to the gene level from mutation positions. It is not clear from the information provided whether the authors distinguish mutation sites that are present in one set of patients, while other mutation sites in the same gene are present in another set of patients within the same cluster. Clarification on this aspect would provide a more comprehensive understanding of their approach.
- During the network learning process, the authors utilize interactions from String for 38 genes. String contains both physical and functional interactions with varying confidence values from different channels. It is unclear from the information provided whether the authors apply any filtering criteria when retrieving interactions. Additionally, the rationale behind specifically seeking direct interactions between the 38 genes needs further explanation, as genes prone to having driver mutations may not necessarily have direct interactions with each other. Elaborating on the selection criteria and the underlying rationale for finding direct interactions would enhance understanding of their approach.
- Are there any mutational signatures associated with each cluster if the authors were to explore COSMIC mutational signatures?
- How do the results obtained from CANClust compare to studies that perform stratification based on diverse omic data types, including transcriptomic profiles? Is CANClust flexible to include other omic data types for stratification?
- My final comment pertains to the robustness and significance of the clusters. Have the authors conducted any randomization tests, such as shuffling mutation profiles across tumors randomly 100 times or employing randomization tests on clinical data, background String interactome, or any other relevant parameters to assess the robustness of their findings?

Reviewer #3

(Remarks to the Author)

In the present study, the authors use a new network-based clustering method to investigate a cohort of 1 325 patients diagnosed with myeloid malignances (AML, MDS, CMML, MPN) at MDAnderson. Using molecular data from a 60-gene

myeloid gene panel, cytogenetic findings, and a restricted set of clinical data, they describe 9 subgroups which cut across the traditional disease entities of myeloid malignancies. They report that the nine "new" subgroups were highly significant in predicting outcome. In addition, for each subgroup the authors study subgroup-specific patterns among mutations, cytogenetic alterations, and clinical patterns and state that these analyses are "offering a more comprehensive understanding of tumor behavior and progression". While well-written, this manuscript fails to significantly advance the field when it comes to arriving at a new molecular classification capable of predicting outcome and to advance our understanding of the interplay between molecular and clinical parameters.

Major criticisms:

1. While the network-based clustering approach taken by the investigators is interesting, it fails to significantly advance current knowledge about the correlation between existing molecular subtypes of the studied disease entities and clinical outcome. For example, the authors identify TP53 mutations to be the most poor-prognostic subtype across the disease entities, referred to as "AML-MDS" (Cluster D). That myeloid malignancies (e.g., AML, MD-related AML, and MDS) with TP53-mutations are associated with an extreme poor prognosis is already well-known. In addition, it is not surprising that Cluster H with *inv(16)* and NPM1-mutations have a moderated-low risk. The same is true for several other clusters as most of them correlated with already known gene mutations, molecular subtypes and/or clinical outcome.
2. The second main results reported by the authors are represented by node-based graphs illustrating the interconnectivity between genetic and/or clinical variables. Again, the reported correlations do not provide any significant new associations. For example, it is well-know that TP53-mutations correlate with a complex karyotype and that NPM1-mutations correlate with FLT3-mutations etc.
3. To validate their network-based clustering approach, the authors used TCGA with a large number of tumor types. It would be more appropriate to validate and benchmark their method to other data sets including myeloid malignancies, e.g., to recent studies in MDS (Bernard et al., 2022, ref 17) and AML (Tazi et al., 2022, ref 15).
4. A final limitation is the design or the possibility to call mutations using the described gene panel: For example, it is well-known that "bi-allelic" TP53-mutations display a worse prognosis than "monoallelic TP53-mutations" in MDS. In addition, the authors do not distinguish between different types of CEBPA-mutations, e.g., only mutations in the bZIP domain of CEBPA are associated with a high degree of remission and better outcome. Similarly, the authors do not report on the association of KMT2A-duplications and "interconnectivity".

Version 1:

Reviewer comments:

Reviewer #3

(Remarks to the Author)

The authors have significantly improved their manuscript. Importantly, as suggested they have validated their approach in two independent and relevant data sets and now more clearly describe the advantages and limitations of their work. A minor remaining issue that caused confusion to this reviewer initially is their use or definition of the term "clinical features", "clinical covariates" etc. For some readers, these terms would include survival data, which was not the case in this manuscript. The authors could consider better clarifying these terms, for example by using "clinical variables at diagnosis" or "clinical variables without taking survival time into account" or similar. In addition, I would recommend the authors to revisit their abstract to more clearly communicate the strengths of their integrative method and its possible extension also to other cancer types.

(Remarks on code availability)

NA

Reviewer #4

(Remarks to the Author)

The manuscript presents a novel approach to classifying myeloid malignancies using network-based clustering that integrates both genomic and clinical data. This method is designed to account for the interplay between different data types and improve upon traditional risk scores. While the approach demonstrates novelty and potential, the clinical relevance remain unclear.

Below are a few major comments:

1. The authors subtract the age effect but do not measure it in the resulting clustering. It would be useful to color the 2D projection in Fig.3 by age to see if any clustering reflects it biologically (as a consequence of the combination of mutation patterns and covariates, not because it is factored in by the model).
2. The main novelty of this model comes from factoring in clinical covariates. It would be useful to stress more their specific

weight in the cluster assignment. This is important because clinical covariates seem to establish fewer probabilistic dependencies compared to genetic variables.

3. The benefit of clustering different myeloid malignancies together should be clearly expressed in the text. While this approach brings together patients with similar genetic properties/risk, it seems to obscure disease-specific characteristics. For example, cluster C is superimposable to cluster D in survival probability for MDS (Figure 4C) but not for AML (Figure 4A). Similarly, cluster A in AML shares similar survival probabilities with cluster C, while better prognosis is only observed for MDS. It would be important to clarify the rationale and implications of clustering all diseases together and discuss how this approach impacts the interpretation of survival probabilities across different diseases.

4. The significance of the positioning and proximity of different clusters in the 2D multidimensional scaling visualisation should be made easier to interpret, for its clinical relevance (if any). When 2D scaled, some clusters remain quite distinct (e.g., cluster D), while others (e.g., cluster A) seem more mixed. It is important to provide interpretability with respect to these observations. For instance, it would be valuable to know if patients in cluster D, who are localised closer to cluster A, display distinct characteristics.

5. Supplementary Figures 2 and 3 are difficult to understand, and the figure descriptions do not sufficiently explain them or their colour legends.

6. Reviewer 1 made an important point by mentioning Zeng et al., work, where gene expression hierarchical signatures were connected to drug response. Of relevance, a well annotated dataset from a preprint from the same lab has recently become available (10.1101/2023.12.26.573390). It would be worthwhile to check and mention this dataset. Even if direct integration is not feasible, an association between mutation patterns and defined phenotypes (and risk) could be discussed in the text.

(Remarks on code availability)

Reviewer #5

(Remarks to the Author)

I co-reviewed this manuscript with one of the reviewers who provided the listed reports. This is part of the Nature Communications initiative to facilitate training in peer review and to provide appropriate recognition for Early Career

(Remarks on code availability)

Reviewer #6

(Remarks to the Author)

As Reviewer #2 is unavailable to review the revised manuscript, I have assessed the authors' responses to the original review and the corresponding changes made to the paper, particularly focusing on the network learning aspect and the specific points raised by Reviewer #2.

The authors propose a network-based clustering approach where genomic and clinical features of tumor samples form an interaction network. Nodes represent features, and edges represent learned interactions. The key assumption is that gene-level mutational features are cluster-dependent (varying across cancer subtypes), while clinical covariates (e.g., age) are cluster-independent. This leads to different clusters having distinct mutational subnetworks within a shared clinical covariate subnetwork. The authors utilize Bayesian network methods (specifically the BiDAG R package) and the EM algorithm to learn the network structure and cluster assignments from the data.

Strengths:

Framework Potential: The network-based framework for clustering tumors holds promise for identifying complex feature interactions and potential subtype-specific network-based biomarkers.

Application to Cancer Genomics: The application of Bayesian networks to analyze cancer genomics data and uncover relationships between mutations and clinical features is a valuable contribution.

Improved Clarity: The authors have successfully addressed Reviewer #2's concerns regarding the clarity of Figure 1 and the justification for including clinical covariates in the network analysis.

Weaknesses:

Limited Algorithmic Novelty: The primary weakness lies in the lack of innovation in computational methodology. The reliance on existing Bayesian network techniques and the EM algorithm, while valid, offers limited novelty in terms of algorithm development.

Scalability Concerns: The algorithm's scalability to larger datasets with tens of thousands of genomic features is a significant concern. The authors acknowledge the computational limitations of their approach, restricting their analysis to a smaller feature set.

Gene-level Resolution: Utilizing gene-level mutational features is a considerable limitation. As Reviewer #3 also points out, different mutations within the same gene can have vastly different effects. In this work, if a selected gene has a mutation (regardless of which mutation it has), the gene-level mutational feature is set to 1 (otherwise 0), which is not accurate. If using fine-grained mutational features (which could easily be in millions) is computational infeasible using the proposed exact algorithm (Algorithm 1), the authors could try to use Ensembl Variant Effect Predictor (VEP) to extract the gene-level mutational features more carefully. It is also not clear how the authors selected the less than 100 gene features in the analysis.

Robustness and Novelty: The authors need to provide more detailed information about the robustness tests performed and their results, addressing Reviewer #2's concerns about the stability of the findings. Furthermore, while the authors highlight the improved prognostic accuracy compared to established risk classifications, a more compelling argument for the unique contributions and novelty of the network learning aspect itself is still needed to address both Review #2 and #3's concerns.

The authors have made commendable efforts to address several of Reviewer #2's concerns. However, the lack of algorithmic novelty, the scalability limitations, the gene-level resolution of features, and the need for stronger evidence of robustness and unique contributions remain significant weaknesses.

Here are some suggestions for potential improvements:

Discussing potential adaptations or modifications to the algorithm to improve scalability and handle higher dimensional data.

Exploring strategies to incorporate finer-grained mutation information (e.g., incorporating mutation impact scores or pathway analysis) to improve the resolution of the network.

Providing a detailed account of the robustness tests performed and their results to demonstrate the stability of their findings.

Clearly articulating the specific novel insights or advantages gained from the network learning aspect beyond the improved prognostic performance.

(Remarks on code availability)

Overall, the code and documentation are well-written. However, I did not run the code, and I am not sure to what extent the results of the paper are reproducible using the code.

Version 2:

Reviewer comments:

Reviewer #4

(Remarks to the Author)

The authors have significantly improved the manuscript and largely addressed my remarks.

(Remarks on code availability)

Reviewer #5

(Remarks to the Author)

(Remarks on code availability)

Reviewer #6

(Remarks to the Author)

The revised manuscript presents a valuable approach for integrating genomic and clinical data to identify cancer subgroups. The findings have clinical relevance in the context of myeloid malignancies. The manuscript could be further improved with a minor revision to address the practical scalability of the method to higher-dimensional datasets.

While theoretical strategies for handling high-dimensional data are discussed, a brief comment on the practical feasibility of applying the current implementation to datasets with thousands of features would be valuable. A rough estimate of computational resource requirements for larger datasets would be helpful.

Overall, the authors have significantly improved the manuscript, addressing concerns about the handling of clinical covariates, the benefits of pan-myeloid clustering, the robustness analysis, and the discussion of limitations. The revised version is stronger, clarifies several important aspects, and provides a more comprehensive analysis. Given the prevalence of both R and Python in bioinformatics, providing a Python implementation of their method alongside the existing R package would likely enhance its adoption, facilitate reproducibility, and enable its integration into diverse analysis pipelines.

(Remarks on code availability)

The GitHub repo of the code is well-structured with a clear README file.

However, I did not install the R package and run the code to reproduce the results presented in the paper.

We thank all the reviewers for their comments which helped us to improve the manuscript, and which we respond to in detail below. We also reorganised the Supplementary Material and updated Figures 2 and 5. The numbering in our responses corresponds to the current version, while changes in the Supplement and the manuscript are highlighted there in blue.

REVIEWER COMMENTS

Reviewer #1 (Remarks to the Author): expertise in AML bioinformatics analysis

Bayer et al introduce a new approach for network-based clustering of cancer patients into groups by mutational and other covariates. They apply their approach to a cohort of 1323 patients diagnosed with various myeloid disorders.

I cannot comment on the mathematics underlying the clustering approach, or its novelty in the machine learning community. My comments take the perspective of an expert on genomics and bioinformatics applications in hematology research.

Superficially, this paper seems a bit like ‘yet another clustering of leukemia patients by mutations’. However, it appears that the new clustering algorithm is well suited for scenarios where both genomic and other covariates are present, and thereby goes beyond the state of the art. In particular, this might have exciting applications beyond the inclusion of simple blood count data, see my 3rd point for a suggestion. It would be good to include such an applications to make this paper interesting to a broader readership.

1. For AML and MPN, the clustering obtained outperforms established risk scores (ELN2022 and IPSS-M) (Figure 5). This might not be a fair comparison, since the risk scores were developed on different cohorts, and still perform well on the 1323 patient cohort analyzed here. By contrast, the clusters were obtained on this specific cohort. To make the point that the clusters can be useful in practice, they would need to do this comparison on an independent cohort.

We thank the reviewer for this comment. Although trained on the patient profiles of the cohort, our clustering does not use any survival information, so predicting survival is still an independent validation on this cohort. The risk scores directly target predicting outcome even if developed on different cohorts, so if anything one might assume they have an advantage by using survival information. We added the following sentence to make this clearer:

“We note that although our cancer subgroups are based on the genomic, clinical and demographic data of the patients, they are not informed by survival data, making this an independent validation, while the established risk scores are based on survival data and target predicting outcome directly.”

Of course, we agree that adding an independent cohort shows generalisability. We therefore included additional analyses, on two new cohorts - one from AML and one from MDS - in a new section called “Classifying New Cohorts with our Novel Subgroups” which indeed supports the utility of our subgroups.

2. The direction of some edges in Figure 2 and 4 creates doubts. For example, DNMT3A mutations generally occur first in leukemia evolution (during clonal hematopoiesis) and is often followed by NPM1, yet the arrow points from NPM1 to DNMT3A. Likewise FLT3 is usually downstream of NPM1.

We appreciate the reviewer's observation regarding the direction of edges in Figures 2 and 5. Indeed, inferring causal or temporal relationships from static observational data, as in our study, poses inherent challenges. The direction of edges in our causal graph may not always indicate causal or temporal relationships, instead they reflect associations observed in the data.

We reworded the figure legend as "Directed edges signify the deduced direction of probabilistic (conditional) dependency, while undirected edges denote instances where this could not be determined." to acknowledge these limitations and better express the meaning of edges in the networks.

3. It could be interesting to compare the clusters obtained here to the cell type based classification presented in Zeng et al (10.1038/s41591-022-01819-x). This paper classified AMLs into a spectrum of "Primitive-to-Mature" and "Primitive-to-GMP" based on RNA-seq. On one side this comparison might shed light on the biological differences between some of the clusters in Figure 3 (e.g. the difference between the NPM1 clusters E and H are not really clear).

On the other hand, the progenitor-based covariates presented in that paper carry higher prognostic value, compared to the dependent covariates currently employed by the authors, and including them into the clustering could further improve the performance here.

In general, RNA-seq based classification of AMLs are becoming increasingly important. The method presented here has the potential for an integrated analysis of mutational data and RNA-seq based features, so including these might take this paper beyond 'yet another clustering of mutations in AML'. If no RNA-seq data is available for the 1323 patients used here, the dataset by Zeng et al could be used.

We appreciate the reviewer's suggestion to compare our clusters with the cell type-based classification presented in Zeng et al. Indeed, such a comparison could offer valuable insights into the biological differences between clusters, particularly where distinctions may not be clear, such as between the NPM1 clusters E and H.

Though integrating progenitor-based covariates may enhance prognostic value, our dataset unfortunately did not include RNA-seq data. We therefore explored the dataset provided by Zeng et al., but found that a significant proportion of the data (~90%) was incomplete, rendering it unsuitable for reliable analysis with our method.

We agree that RNA-seq-based classifications are becoming increasingly important in AML research, and our method does hold promise for integrating mutational data with RNA-seq features. We added the following to the discussion (also in response to referee #2) to emphasise the future direction of collecting and integrating multi-modal data:

"While our investigations predominantly considered bulk DNA sequencing data, cytogenetic features, and clinical attributes, the leukemia cellular hierarchy has also been shown to

associate with genomic and clinical properties [45] and integrating our approach with additional data facets, like such transcriptomic data, could further refine the subgroups and enhance their prognostic value. Handling multi-omic data in network learning has previously been considered [25] and could be combined with our method to integrate clinical covariates”

4. The documentation of the accompanying R package needs to be improved, e.g. the vignette was empty when I tried to open it

We thank the reviewer for this comment and we’re sorry for this oversight. The documentation has been updated and the vignette added, and can be accessed via: <https://cran.r-project.org/web/packages/clustNet/vignettes/clustNet.html>

5. Figure 2&4: Abbreviations are not defined and the directionality of the arrows is hard to see.

We thank the reviewer for pointing this out and made the following changes to the manuscript: First, we changed the abbreviations of the cytogenetic factors such that they are more clear. Second, we changed the size of the arrows so that the directionality becomes better visible.

6. Some statements in the discussion (e.g. use of AML therapies in MDS) should be commented on by a reviewer with a clinical expertise, or removed.

Thanks for raising this - we (and specifically the clinical co-authors among us) reworked this and expanded it with references to the following:

“Since MDS is a heterogeneous disease, this aligns with the general observation that higher-risk MDS behaves more similarly to AML including for TP53-altered subtypes [42]. Historically higher-risk MDS has also been treated with AML-like regimens [43,44], and ongoing clinical trials are exploring the use of AML-approved treatments in higher-risk MDS (VERONA trial: NCT04401748). Our clustering might then help predict the subset of patients who might benefit from AML-like therapies.”

Reviewer #2 (Remarks to the Author): expert in network learning and visualisation

Bayer et al present a data-driven network-based clustering approach that integrates mutational features and clinical covariates to discover de novo cancer subgroups. They apply their approach to a cohort of 1323 patients to identify novel subgroups. Based on their results, this method outperforms established risk classifications in prognostic accuracy. They also apply this method to TCGA data to show differences in the survivals of patient groups. The manuscript raises an important topic however, I have stills some major comments that need to be addressed.

- The clarity of Figure 1 is limited, as it lacks self-description. While I recognize its abstract nature intended to convey the concept, the right panel, particularly the cluster labels, is challenging to decipher. A comprehensive revision and elaboration of the figure legend are essential to enhance understanding.

We thank the reviewer for suggesting this and completely reworded the caption to:

“The data of patients (left panels) across different cancer types (colouring on the left) include mutational profiles (X_1 , X_2 , ..., top left) and clinical covariates (C_1 , C_2 , ..., bottom left). We model the dependencies amongst the variables in the data with probabilistic network models (right) and use these to cluster the patients into new subgroups based on their mutation profiles and distinct probabilistic relationships between the mutations (solid edges, right). Since mutations may depend on the clinical covariates, we account for these dependencies in the network modelling (dashed edges, top right) allowing us to adjust for them in the cluster assignments. The result of the network-based clustering are different network models (shades of blue, right panel) and the assignment of patients to the clusters (bottom right).”

- While the authors assert in the text that 'To avoid learning the clusters based on demographic covariates such as age and sex, we remove their effect on the cluster assignment by exploiting causal relationships among the variables,' notably, these covariates are still present in the networks depicted in Figure 2. Is it necessary to have them in the networks? If yes, why?

As in Figure 1, including these variables in the network is necessary to better learn the dependencies between the other variables. If we exclude them the downstream relationships would be confounded and potentially learned incorrectly. The effect of these variables is instead adjusted for when we cluster. As the reviewer mentions, the word “remove” was not a good choice for describing how we handle the variables and we have replaced it with “adjust for”.

- If the authors were to apply a naive method, clustering tumors based solely on mutational profiles and clinical covariates, without considering causal relationships through a standard clustering method, would the resulting clusters significantly overlap or not with theirs? This analysis could provide valuable insights into the distinct contributions of their proposed method to patient stratification.

We appreciate the insightful comment from the reviewer. We previously considered this in the supplementary analysis provided in section C, where we demonstrate that incorporating clinical covariates significantly enhances the accuracy of survival prediction compared to utilizing genomic information alone (Supplementary Table S8). We have now also checked this for the pan-myeloid cohort with the result in Supplementary Table S4 and added the following text:

“With the clustering we can see the importance of including and appropriately adjusting for the clinical covariates. Without adjustment our survival predictions are worse, while they worsen further when considering mutations only (Supplementary Table S4)”

- In the cancer-specific and cluster networks, edges solely represent the direction of causal relationships. Nevertheless, in the text, the authors also refer to the mutual exclusivity and co-occurrence of genomic alterations, such as JAK2, CALR, and MPL mutations, as well as JAK2 and del(20q) in MPN. Do they have this specific information? If yes, this can be represented in the networks with different colors of edges.

We thank the reviewer for this great suggestion. We have updated Figure 2 and Figure 5 (and their corresponding supplementary Figures) to display positive correlation via solid edges and negative correlation via dashed edges.

- In 2D projection for visualization purposes in Figure 3a, which algorithm has been selected? If that technique were changed, still the same representation be obtained?

Sorry we didn't specify - we used multidimensional scaling and have added this to the caption. The 2D projection would change a lot if other visualisation techniques like tSNE or UMAP were employed, which in turn depend heavily on parameter choices. We preferred multidimensional scaling as it offers the “best” representation of the original distances in terms of the squared differences in true and projected distances, and does not have any further parameters to tweak.

- Each cluster possesses unique properties and significant pairs within their networks. Maintaining a consistent layout and restricting nodes to the most frequent and highly interconnected variables across all clusters results in sparse cluster-specific networks. I recommend visualizing the clusters by limiting their edges rather than nodes. This approach ensures that the most crucial interactions are highlighted in the main text, and those causal relations specific to clusters are shown in the main figure.

We thank the reviewer for this suggestion. To address this, we selected nodes more by the connectedness of the edges than by frequency in Figure 2 and Figure 5.

- Regarding the TCGA analysis, they utilized pan-cancer data, including solid tumors. Although the method is originally designed for blood cancers, the rationale behind extending it to pan-cancer data needs further clarification.

The clustering method is more general, but the referee raises a good point that we did not clarify the differences carefully and how our framework can cover different modelling choices. In fact, for the TCGA, the mutation profiles will depend on the tissue of origin, and

we therefore also adjust for that variable in the clustering. For the pan-myeloid data instead, the genomic data can determine the cancer type, so we do not adjust for it. We have modified the supplement in the TCGA analysis to emphasise this as:

“Since the variables age, sex and tissue of origin have a downstream causal effect on the mutations, we adjusted for them in our covariate-adjusted clustering framework. This is different from our modelling of the pan-myeloid data where the cancer type can be a consequence of the genomic data and is hence excluded from the adjustment.”

- Within each cluster of tumors, the authors reduce the data to the gene level from mutation positions. It is not clear from the information provided whether the authors distinguish mutation sites that are present in one set of patients, while other mutation sites in the same gene are present in another set of patients within the same cluster. Clarification on this aspect would provide a more comprehensive understanding of their approach.

We thank the reviewer for this clarification. The mutation data considered was binary, indicating only the presence or absence of mutations, summarised at the gene level, without distinguishing the exact position of the mutation. We added the following to the revised text to clarify:

“Mutations were summarised at the gene level and recorded as binary values indicating the presence of any mutation in that gene or their absence.”

In principle though one can go to higher granularity and consider point mutations, at the cost of higher dimensions and sparser data, which affects the ease and accuracy of network learning. This trade-off is therefore a modelling choice. To reference this point better, we added the following to the discussion (also in response to referee #3):

“In our analysis, we summarised mutations at the gene level, though different point mutations in the same gene may behave differently. Increasing the granularity, and resolving mutations down to individual genomic positions, likewise increases the dimensionality of the network, along with the sparsity of the data and the complexity of network learning. For certain mutations, however, accounting for their potentially different effects in the same gene may still increase predictive accuracy.”

- During the network learning process, the authors utilize interactions from String for 38 genes. String contains both physical and functional interactions with varying confidence values from different channels. It is unclear from the information provided whether the authors apply any filtering criteria when retrieving interactions. Additionally, the rationale behind specifically seeking direct interactions between the 38 genes needs further explanation, as genes prone to having driver mutations may not necessarily have direct interactions with each other. Elaborating on the selection criteria and the underlying rationale for finding direct interactions would enhance understanding of their approach.

Sorry we did not specify - we filtered for known human functional interactions, and have updated the text accordingly. We only used the database as a prior weight on edge presence, so our network learning can still learn edges outside, we just need more evidence from the data to do so. The rationale for using this as prior information is that we hope edges

known in STRING would be more relevant. We did check however that using the STRING database in this way to guide the network learning seems to help, since without it we perform slightly worse in the survival prediction. We included the following about this:

“Likewise using prior knowledge from the STRING database (Methods) aids the network learning as excluding this also leads to worse survival prediction (Supplementary Table S4 last row) with changes in cluster assignment shown in Supplementary Figure S5.”

- Are there any mutational signatures associated with each cluster if the authors were to explore COSMIC mutational signatures?

Our data is limited to a panel of profiled genes, which would not be sufficient to explore the COSMIC signatures since they are based on WES or WGS.

- How do the results obtained from CANClust compare to studies that perform stratification based on diverse omic data types, including transcriptomic profiles? Is CANClust flexible to include other omic data types for stratification?

For now we focussed on the genetic and clinical data that may be routinely collected, but agree that integrating further data types would be an interesting avenue. We added the following to the discussion (also in response to referee #1) to touch on these directions:

“While our investigations predominantly considered bulk DNA sequencing data, cytogenetic features, and clinical attributes, the leukemia cellular hierarchy has also been shown to associate with genomic and clinical properties [45] and integrating our approach with additional data facets, like such transcriptomic data, could further refine the subgroups and enhance their prognostic value. Handling multi-omic data in network learning has previously been considered [25] and could be combined with our method to integrate clinical covariates”

- My final comment pertains to the robustness and significance of the clusters. Have the authors conducted any randomization tests, such as shuffling mutation profiles across tumors randomly 100 times or employing randomization tests on clinical data, background String interactome, or any other relevant parameters to assess the robustness of their findings?

We thank the reviewer for this comment. Along with changes mentioned above, we did several things to address this:

- 1. Robustness Assessment: As suggested, we conducted randomization tests by rerunning with different random starting points and random seeds (akin to shuffling the patients) 100 times. We then calculated the adjusted rand index (ARI) to measure the similarity across the clusters. The high ARI values obtained (mean ARI = 0.92, Supplementary Figure S10) show the robustness of the clustering, and that there is some small amount of stochastic variation. We added the small section “Robustness” to the Methods to include these results.*

2. *Additional Analysis: As above we looked at the effect of the STRING prior and (also in response to the other referees) we included analyses of independent cohorts, described in the new “Classifying New Cohorts with our Novel Subgroups” section which aligns with the advantages found for our cohort and points to the generalizability of our results.*
3. *Supplementary Material: This already contains benchmarks on simulated data and additional analyses on TCGA data. These supplementary analyses also show generally robust out-performance over other methods and the clinical significance of our method.*

Reviewer #3 (Remarks to the Author): clinical expertise in AML genomics

In the present study, the authors use a new network-based clustering method to investigate a cohort of 1 325 patients diagnosed with myeloid malignancies (AML, MDS, CMML, MPN) at MDAnderson. Using molecular data from a 60-gene myeloid gene panel, cytogenetic findings, and a restricted set of clinical data, they describe 9 subgroups which cut across the traditional disease entities of myeloid malignancies. They report that the nine “new” subgroups were highly significant in predicting outcome. In addition, for each subgroup the authors study subgroup-specific patterns among mutations, cytogenetic alterations, and clinical patterns and state that these analyses are “offering a more comprehensive understanding of tumor behavior and progression”. While well-written, this manuscript fails to significantly advance the field when it comes to arriving at a new molecular classification capable of predicting outcome and to advance our understanding of the interplay between molecular and clinical parameters.

Major criticisms:

1. While the network-based clustering approach taken by the investigators is interesting, it fails to significantly advance current knowledge about the correlation between existing molecular subtypes of the studied disease entities and clinical outcome. For example, the authors identify TP53 mutations to be the most poor-prognostic subtype across the disease entities, referred to as “AML-MDS” (Cluster D). That myeloid malignancies (e.g., AML, MD-related AML, and MDS) with TP53-mutations are associated with an extreme poor prognosis is already well-known. In addition, it is not surprising that Cluster H with inv(16) and NPM1-mutations have a moderated-low risk. The same is true for several other clusters as most of them correlated with already known gene mutations, molecular subtypes and/or clinical outcome.

Thank you for your comment! By taking the mutational and cytogenetic profiles, and accounting appropriately for clinical covariates, with an unsupervised analysis we have indeed recapitulated a lot of known knowledge. We stress that this is without having to put the knowledge in in the first place. The fact the features align well with current knowledge is a robust sign that we are learning a meaningful representation of the data. The key point however is that our clustering is not based on single features like a particular mutation, but a holistic treatment of all the data. The fact that our clustering is a stronger predictor of survival than the current risk scores (which focus on a small number of individual features), is testament to the relevance of treating such data holistically. We added the following to the discussion to emphasise this:

“Unlike the risk scores which focus on a few low-dimensional features predictive of outcome, our network-based approach treats the entirety of the data holistically including its complex interdependencies also with clinical covariates. This allows it to better model the data, and even predict survival better despite being trained independently without access to the survival data.”

2. The second main results reported by the authors are represented by node-based graphs illustrating the interconnectivity between genetic and/or clinical variables. Again, the reported correlations do not provide any significant new associations. For example, it is well-know

that TP53-mutations correlate with a complex karyotype and that NPM1-mutations correlate with FLT3-mutations etc.

Again it is not the individual edges which are necessarily new, but the treatment of the complete high-dimensional data, and its internal dependencies.

3. To validate their network-based clustering approach, the authors used TCGA with a large number of tumor types. It would be more appropriate to validate and benchmark their method to other data sets including myeloid malignancies, e.g., to recent studies in MDS (Bernard et al., 2022, ref 17) and AML (Tazi et al., 2022, ref 15).

We thank the reviewer for this comment and agree that independent cohorts add considerably to the validation. To address this, we added an additional analysis of validation cohorts (see section “Classifying New Cohorts with our Novel Subgroups” and Figure S8, and Table S7). There we again find an improvement in survival prediction over the risk score and see the advantages and generalisability of our approach in classifying new populations.

4. A final limitation is the design or the possibility to call mutations using the described gene panel: For example, it is well-known that “bi-allelic” TP53-mutations display a worse prognosis than “monoallelic TP53-mutations” in MDS. In addition, the authors do not distinguish between different types of CEBPA-mutations, e.g., only mutations in the bZIP domain of CEBPA are associated with a high degree of remission and better outcome. Similarly, the authors do not report on the association of KMT2A-duplications and “interconnectivity”.

We agree that this is a limitation, or rather a trade-off. The finer the resolution we move to in the gene-point mutation spectrum, the higher dimensional and sparser the data becomes. It would be possible to have different resolutions for different mutations, for example the ones the referee mentions, to better account for their different behaviours. To reference this point better, we added the following to the discussion (also in response to referee #2):

“In our analysis, we summarised mutations at the gene level, though different point mutations in the same gene may behave differently. Increasing the granularity, and resolving mutations down to individual genomic positions, likewise increases the dimensionality of the network, along with the sparsity of the data, and the complexity of network learning. For certain mutations, however, accounting for their potentially different effects in the same gene may still increase predictive accuracy.”

We thank the Editor and Reviewers for their comments and suggestions, which we respond to inline below in blue italic text.

Reviewer #3 (Remarks to the Author):

The authors have significantly improved their manuscript. Importantly, as suggested they have validated their approach in two independent and relevant data sets and now more clearly describe the advantages and limitations of their work. A minor remaining issue that caused confusion to this reviewer initially is their use or definition of the term "clinical features", "clinical covariates" etc. For some readers, these terms would include survival data, which was not the case in this manuscript. The authors could consider better clarifying these terms, for example by using "clinical variables at diagnosis" or "clinical variables without taking survival time into account" or similar. In addition, I would recommend the authors to revisit their abstract to more clearly communicate the strengths of their integrative method and its possible extension also to other cancer types.

Thanks for raising this point, which we had not appreciated, and the suggestion to clarify. We have added "at diagnosis" to the first mentions of clinical covariates/features in the abstract and introduction and added the following sentence to the last paragraph of the introduction to make clear:

"Throughout, we include only baseline covariates among the clinical features and exclude survival information, which we instead predict to validate our approach."

Thanks also for the suggestion to improve the abstract. We added the following sentences throughout the abstract to emphasise the strengths and wider applicability:

"A key strength of the approach is its ability to include presumed causal directions in the edges linking clinical and mutational features, and to account for them aptly in the clustering."

"Our approach generalises well to new cohorts with classification based on our subgroups similarly offering advantages in predicting prognosis."

"Furthermore, with pan-cancer TCGA data, we observe that our modelling and clustering framework extends naturally to other cancer types while continuing to offer improvements in subgroup discovery."

Reviewer #3 (Remarks on code availability):

NA

Reviewer #4 (Remarks to the Author):

The manuscript presents a novel approach to classifying myeloid malignancies using network-based clustering that integrates both genomic and clinical data. This method is designed to account for the interplay between different data types and improve upon traditional risk scores. While the approach demonstrates novelty and potential, the clinical relevance remain unclear.

Thanks for the summary, for the clinical relevance we argue that offering better prognosis prediction, also for new validation cohorts, indicates that we are uncovering clinically relevant subgroups, while we also observe connections in a spectrum between MDS and AML which touches on one of the points below.

Below are a few major comments:

1. The authors subtract the age effect but do not measure it in the resulting clustering. It would be useful to color the 2D projection in Fig.3 by age to see if any clustering reflects it biologically (as a consequence of the combination of mutation patterns and covariates, not because it is factored in by the model).

Thanks for this observation. Although we condition on age in the cluster assignment, because it may influence other variables and their connections, it is correct that we may still expect different clusters to display different age distributions. To highlight this possibility, we have made a version of Fig 3 coloured by age and included it as a new Supplementary Figure S3 where we also show the age distribution per cluster. Since the age distributions are not identical, it means that while conditioning on age avoids a direct effect on the clustering, it still captures indirect effects reflected in the network structure among other variables. We added the following to the manuscript to touch on these points:

“Though we condition on age in the clustering assignment, since it may influence other variables and their connections, different age distributions across clusters may be expected, reflecting their different compositions in terms of cancer types as well as mutations. The distributions largely overlap since the conditioning ensures that the clustering is not driven by age itself (Supplementary Figure S3).”

2. The main novelty of this model comes from factoring in clinical covariates. It would be useful to stress more their specific weight in the cluster assignment. This is important because clinical covariates seem to establish fewer probabilistic dependencies compared to genetic variables.

Cluster assignment is conditional on clinical covariates, especially to uncover independent information available through genetic variables. While they may be prognostic of survival (a reason why we wish to condition on them), their influence on genetic features is less obvious, and the more limited number of probabilistic dependencies is not necessarily surprising. What is important is that by conditioning on them, their direct effect on the clustering is discounted, while

still allowed indirectly through their effect on other variables and their connections (see also response to previous point).

3. The benefit of clustering different myeloid malignancies together should be clearly expressed in the text. While this approach brings together patients with similar genetic properties/risk, it seems to obscure disease-specific characteristics. For example, cluster C is superimposable to cluster D in survival probability for MDS (Figure 4C) but not for AML (Figure 4A). Similarly, cluster A in AML shares similar survival probabilities with cluster C, while better prognosis is only observed for MDS. It would be important to clarify the rationale and implications of clustering all diseases together and discuss how this approach impacts the interpretation of survival probabilities across different diseases.

We indeed noticed that, along with differences between the different diseases, there were similarities, and we appreciate the referee's correct observation that we did not explicitly test whether treating the cohorts together was better. To address this point, we have added another analysis where we compare treating the AML and MDS patients together and learning the separations through clustering, to the case where we stratify beforehand and treat the two entities separately. The combined analysis has better performance, and we added the following to the manuscript:

"We can additionally see the benefit of clustering in a pan-myeloid setting. For example, if we cluster the AML and MDS patients together we obtain better predictions than treating the two separately (Supplementary Table S5)."

4. The significance of the positioning and proximity of different clusters in the 2D multidimensional scaling visualisation should be made easier to interpret, for its clinical relevance (if any). When 2D scaled, some clusters remain quite distinct (e.g., cluster D), while others (e.g., cluster A) seem more mixed. It is important to provide interpretability with respect to these observations. For instance, it would be valuable to know if patients in cluster D, who are localised closer to cluster A, display distinct characteristics.

We chose 2D multidimensional scaling since it preserves the original distances better than t-SNE or UMAP projections, so closer points generally correspond to more similar patient profiles. However, since it is a projection to lower dimensions there is a loss and the correspondence is not exact. We added the following to the manuscript:

"The projection employed (multi-dimensional scaling) preserves distances as well as possible given the need to reduce to two dimensions, so closer points generally correspond to more similar patient profiles, though the correspondence is not exact."

For cluster D, when we split them by their weight to cluster A we indeed observe a difference in survival, with those most similar to A having better survival. We included this analysis as a new Supplementary Figure (S5) and added the following to the manuscript to discuss this aspect:

“Though the clustering is a discretisation of each patient’s assigned weight to the different clusters, the continuous weights (reflected in the positioning in the two-dimensional projection, Figure 3a,b) may be informative too. For example, if we stratify patients in cluster D by their similarity to cluster A, we find a significant difference (Supplementary Figure S5).”

5. Supplementary Figures 2 and 3 are difficult to understand, and the figure descriptions do not sufficiently explain them or their colour legends.

Thanks for raising this, and we agree that they were not adding to the work. We have replaced Supplementary Figure S2 with enlarged versions of panels of Figure 2 to make them clearer and replaced Supplementary Figure S3 with the age distributions.

6. Reviewer 1 made an important point by mentioning Zeng et al., work, where gene expression hierarchical signatures were connected to drug response. Of relevance, a well annotated dataset from a preprint from the same lab has recently become available (10.1101/2023.12.26.573390). It would be worthwhile to check and mention this dataset. Even if direct integration is not feasible, an association between mutation patterns and defined phenotypes (and risk) could be discussed in the text.

Thanks for pointing out this follow-on work, especially the link between the signatures and subclones. We agree that considering subpopulations of cells in each cancer could be highly relevant and added the following to the discussion to reference this:

“Moreover, different transcriptomic programs can co-exist within the same cancer and be linked to different genetic subclones [Zeng et al 2023], while single-cell DNA sequencing has further illuminated the wealth of subclones and evolutionary patterns in AML, along with their links to phenotypic heterogeneity [Schwede et al 2024]. These results suggest that in addition to data integration, treating cancers at the subclonal level could also lead to better modelling of subgroups and prognosis”.

Reviewer #5 (Remarks to the Author):

I co-reviewed this manuscript with one of the reviewers who provided the listed reports. This is part of the Nature Communications initiative to facilitate training in peer review and to provide appropriate recognition for Early Career

Reviewer #6 (Remarks to the Author):

As Reviewer #2 is unavailable to review the revised manuscript, I have assessed the authors' responses to the original review and the corresponding changes made to the paper, particularly focusing on the network learning aspect and the specific points raised by Reviewer #2.

The authors propose a network-based clustering approach where genomic and clinical features of tumor samples form an interaction network. Nodes represent features, and edges represent learned interactions. The key assumption is that gene-level mutational features are cluster-dependent (varying across cancer subtypes), while clinical covariates (e.g., age) are cluster-independent. This leads to different clusters having distinct mutational subnetworks within a shared clinical covariate subnetwork. The authors utilize Bayesian network methods (specifically the BiDAG R package) and the EM algorithm to learn the network structure and cluster assignments from the data.

Strengths:

Framework Potential: The network-based framework for clustering tumors holds promise for identifying complex feature interactions and potential subtype-specific network-based biomarkers.

Application to Cancer Genomics: The application of Bayesian networks to analyze cancer genomics data and uncover relationships between mutations and clinical features is a valuable contribution.

Improved Clarity: The authors have successfully addressed Reviewer #2's concerns regarding the clarity of Figure 1 and the justification for including clinical covariates in the network analysis.

Thanks for stepping in to evaluate the manuscript and our previous responses, and for the above summary.

Weaknesses:

Limited Algorithmic Novelty: The primary weakness lies in the lack of innovation in computational methodology. The reliance on existing Bayesian network techniques and the EM algorithm, while valid, offers limited novelty in terms of algorithm development.

We would like to stress that the importance of the contribution lies not only in the algorithms we use but also in how we handle and model the data. Here we show that integrating clinical covariates with an approach leaning into the causally-related directionality properties of Bayesian networks enables us to extract much more useful information from the data, for example, finding better and more highly prognostic subgroups. Bayesian network learning is itself a broad field (which we are quite active in developing and also in comparing algorithms, see for example benchpressdocs.readthedocs.io), and with this study we demonstrate the

practical relevance of state-of-the-art techniques. We consider it important to recognise that, beyond algorithmic developments, sub-optimal data modelling will impact the quality of data analysis and principled epidemiological modelling of data will remain key.

Scalability Concerns: The algorithm's scalability to larger datasets with tens of thousands of genomic features is a significant concern. The authors acknowledge the computational limitations of their approach, restricting their analysis to a smaller feature set.

For heterogeneous cancers and patients, even with thousands of samples, there is no way to have a model with tens of thousands of features - the sheer number of parameters would vastly outweigh what can be learned from the data. Either one needs to perform dimensionality reduction to lower dimensions or perform feature selection. For the myeloid malignancies we chose clinically relevant parameters and genes and had a relatively small feature set. This was selected based on clinical knowledge, not on the algorithmic limitations since we had dozens of variables which is well below the hundreds (like in the TCGA example) where the network inference starts to get more computationally demanding.

Gene-level Resolution: Utilizing gene-level mutational features is a considerable limitation. As Reviewer #3 also points out, different mutations within the same gene can have vastly different effects. In this work, if a selected gene has a mutation (regardless of which mutation it has), the gene-level mutational feature is set to 1 (otherwise 0), which is not accurate. If using fine-grained mutational features (which could easily be in millions) is computationally infeasible using the proposed exact algorithm (Algorithm 1), the authors could try to use Ensembl Variant Effect Predictor (VEP) to extract the gene-level mutational features more carefully. It is also not clear how the authors selected the less than 100 gene features in the analysis.

We added to the discussion this and other ways of summarising genes, with the text:

“One way to collate this to summarise at the gene level is to combine the predicted pathogenicity of each mutation, for example with the Ensembl variant effect predictor [McClaren et al, 2016] which includes the SIFT [Ng & Henikoff, 2001] and PolyPhen [Adzhubei et al, 2010] predictors. More recently, the effects of mutations have been summarised through neural network models trained on cell-line drug-response data [Wall & Ideker, 2024]. These types of continuous summaries could then be used to replace binarised values in analyses such as ours. Alternatively, mutations could be combined at the pathway level, which could also reduce the dimensionality of the networks.”

For the gene selection, the patients at MDA undergo targeted sequencing for a panel of 81-genes (not the 60 we previously stated) and global karyotyping, and we now include a reference to this panel. The features were then filtered to those with a prevalence above 1% to the final set of features. We have corrected and updated the text so these details are now included as:

“Cytogenetics was performed by global karyotyping of each sample, while mutations were detected through an 81-gene targeted sequencing panel based on recurrent somatic mutations in myeloid malignancies [Sperling et al, 2022], which form part of the clinical sequencing used at MD Anderson for clinical decision making. ... Features with a prevalence below 1% were excluded, narrowing our focus to 38 genes and eight cytogenetic profiles.”

Robustness and Novelty: The authors need to provide more detailed information about the robustness tests performed and their results, addressing Reviewer #2's concerns about the stability of the findings. Furthermore, while the authors highlight the improved prognostic accuracy compared to established risk classifications, a more compelling argument for the unique contributions and novelty of the network learning aspect itself is still needed to address both Review #2 and #3's concerns. The authors have made commendable efforts to address several of Reviewer #2's concerns.

Our previous revision addressed Reviewer #3's concerns above, apart from those arising due to our misunderstanding of different definitions of what constitutes clinical data, which we have now also addressed. As detailed above, we have now made this clear throughout the manuscript and reworded the abstract to make the unique contribution clearer.

We previously provided a detailed response to all the points raised by Reviewer #2. Further, the new clarifications above about the role of age (point 1 of Reviewer #4), along with the analyses comparing our causal framework to approaches without such modelling which show the advantages of our approach, also address some previous concerns of Reviewer #2.

However, the lack of algorithmic novelty, the scalability limitations, the gene-level resolution of features, and the need for stronger evidence of robustness and unique contributions remain significant weaknesses.

Here are some suggestions for potential improvements:

Discussing potential adaptations or modifications to the algorithm to improve scalability and handle higher dimensional data.

We added to the discussion the following points about how one could reduce the difficulties in network learning:

“One approach to reducing these demands is to enforce sparsity through regularising the network to focus on the strongest connections. With sufficient regularisation, the network learning can be divided into smaller sub-problems and even reduced to polynomial complexity [Rios et al, 2024].”

“Alternatively, mutations could be combined at the pathway level, which could also reduce the dimensionality of the networks.”

Exploring strategies to incorporate finer-grained mutation information (e.g., incorporating mutation impact scores or pathway analysis) to improve the resolution of the network.

As mentioned above, we enriched the discussion with the following to touch on these points:

“One way to collate this to summarise at the gene level is to combine the predicted pathogenicity of each mutation, for example with the Ensembl variant effect predictor [McClaren et al, 2016] which includes the SIFT [Ng & Henikoff, 2001] and PolyPhen [Adzhubei et al, 2010] predictors. More recently, the effects of mutations have been summarised through neural network models trained on cell-line drug-response data [Wall & Ideker, 2024]. These types of continuous summaries could then be used to replace binarised values in analyses such as ours. Alternatively, mutations could be combined at the pathway level, which could also reduce the dimensionality of the networks.”

However there are many ways of exploring these directions, and a full account would seem more fitting for follow-up work. For any of these strategies, the causally-inspired modelling of accounting for covariates in a principled way would remain a key component of such analyses, since, as we showed with this manuscript, doing so consistently improves the results for the binarised data.

Providing a detailed account of the robustness tests performed and their results to demonstrate the stability of their findings.

We have extended our previous description, including the exact details of how the tests were performed and updated the manuscript to include this information as:

“Clustering high-dimensional data involves exploring the large space of possible patient groupings, which may be challenging and contain many local optima. We therefore wished to check the robustness of our clustering procedure. As assigning an equal weight to each patient is an unstable equilibrium clustering solution, small perturbations can lead to the clustering algorithm initially exploring diverse parts of the clustering space. We then check the robustness of the final cluster assignments.

In particular, we ran the entire clustering algorithm 100 times from different starting perturbations. For the starting point, from an equal weight per patient, we increased the relative weight by 0.1% independently for each patient to a uniformly sampled cluster. As the measure of robustness we computed the adjusted Rand index (ARI) of the different clustering outcomes compared to the clustering reported in the main text.”

Clearly articulating the specific novel insights or advantages gained from the network learning aspect beyond the improved prognostic performance.

As in response to Reviewer #3, we added the following sentences throughout the abstract to emphasise the strengths and wider applicability:

“A key strength of the approach is its ability to include presumed causal directions in the edges linking clinical and mutational features, and to account for them aptly in the clustering.”

“Our approach generalises well to new cohorts with classification based on our subgroups similarly offering advantages in predicting prognosis.”

“Furthermore, with pancancer TCGA data, we observe that our modelling and clustering framework extends naturally to other cancer types while continuing to offer improvements in subgroup discovery.”

Reviewer #6 (Remarks on code availability):

Overall, the code and documentation are well-written. However, I did not run the code, and I am not sure to what extent the results of the paper are reproducible using the code.

We thank the Editor and Reviewers for their remaining comments and suggestions, which we respond to inline below in blue italic text.

Reviewer #6 (Remarks to the Author):

The revised manuscript presents a valuable approach for integrating genomic and clinical data to identify cancer subgroups. The findings have clinical relevance in the context of myeloid malignancies. The manuscript could be further improved with a minor revision to address the practical scalability of the method to higher-dimensional datasets.

While theoretical strategies for handling high-dimensional data are discussed, a brief comment on the practical feasibility of applying the current implementation to datasets with thousands of features would be valuable. A rough estimate of computational resource requirements for larger datasets would be helpful.

We agree that discussing the runtimes is important for practical feasibility, and we had previously included this in the Supplementary Material in Section A.2, and now include this also in the discussion at the start of the seventh paragraph so it now reads:

“Although we incorporate tens to hundreds of features across thousands of patients, the computational cost grows with the number of features. For example the clustering takes minutes for small examples with 20 features, several hours for the pan-myeloid data with around 60 and just over a day for the TCGA data with 200 (see Supplementary Section A.2). The scenario could then become ...”

Overall, the authors have significantly improved the manuscript, addressing concerns about the handling of clinical covariates, the benefits of pan-myeloid clustering, the robustness analysis, and the discussion of limitations. The revised version is stronger, clarifies several important aspects, and provides a more comprehensive analysis. Given the prevalence of both R and Python in bioinformatics, providing a Python implementation of their method alongside the existing R package would likely enhance its adoption, facilitate reproducibility, and enable its integration into diverse analysis pipelines.

Thanks for this summary, and we'll keep in mind the suggestion for a Python package too.